# Quantitative Ethnobotany of Medicinal Plants from Darjeeling District of West Bengal, India, along with Phytochemistry and Toxicity Study of *Betula alnoides* Buch.-Ham. ex D.Don bark

**DOI:** 10.3390/plants13243505

**Published:** 2024-12-16

**Authors:** Yasodha Subba, Samik Hazra, Chowdhury Habibur Rahaman

**Affiliations:** Ethnopharmacology Laboratory, Department of Botany, Visva-Bharati University, Santiniketan 731235, West Bengal, India; 03333321910@visva-bharati.ac.in (Y.S.);

**Keywords:** ethnomedicine, Darjeeling Himalaya, ethnobotanical indices, bark extract, HPLC, MTT assay

## Abstract

This study offers considerable information on plant wealth of therapeutic importance used traditionally by the residents of 11 villages under three subdivisions of Kurseong, Darjeeling Sadar, and Mirik in the Darjeeling District, West Bengal. For the acquisition of ethnomedicinal information, semi-structured interviews were conducted with 47 informants, of whom 11 persons were herbalists and 36 were knowledgeable persons. Free prior informed consent was obtained from each participant prior to the collection of field data. A total of 115 species were documented, which spread over 65 families and 104 genera. From the informants, a total of 101 monoherbal and 21 polyherbal formulations were recorded for treating 50 types of health conditions. The collected ethnobotanical data have been evaluated to measure the utilitarian significance of remedies using three quantitative tools, informant consensus factor (F_ic_), use value (UV), and fidelity level (FL%). A statistical analysis revealed that among 11 disease categories, the highest F_ic_ value was estimated for the category of digestive diseases. The plant *Hellenia speciosa* (J.Koenig) S.R.Dutta scored the highest use value among all the recorded plant species. In the case of the FL% analysis, the highest score (97%) was observed in *Betula alnoides* Buch-Ham. ex D.Don, which is used for snake bites, among the recorded 115 plant species. In addition, the present study embodies the quantitative estimation of phenolics and flavonoids, along with an HPLC analysis of the *B*. *alnoides* bark to endorse this most important and underexplored plant as a potential source of therapeutically important chemical compounds. The bark extract contains significant amounts of phenolics (87.8 mg GAE/g dry tissue) and flavonoids (30.1 mg CE/g dry tissue). An HPLC analysis unveiled a captivating ensemble of six phenolic compounds, namely, chlorogenic acid, sinapic acid, caffeic acid, coumarin, p-coumaric acid, and gallic acid. Among the identified phenolics, chlorogenic acid scored the highest amount of 117.5 mg/g of dry tissue. The present study also explored the moderate cytotoxic nature of the bark extract through an in vitro cytotoxicity assay on the L929 mouse fibroblast cell line. Our study not only documents the statistically analyzed information about ethnomedicinal practices that prevailed in the rural communities of the Darjeeling District but also highlights the profound therapeutic capabilities and non-toxic nature of *B*. *alnoides* bark.

## 1. Introduction

Plant-based products have been recognized as a fundamental component in the field of medicine since time immemorial. Plants with therapeutic attributes possess phytochemicals in their different parts that are responsible for healing ailments. The chemical compounds present in the plant materials elicit or alter the specific physiological responses in the ailed person and thus facilitate the treatment of various diseases both in humans and animals [1,2]. In traditional medicine, the diversity and abundance of medicinal plant resources are determined not only by the richness of the regional flora but also by the availability and convenience of using such plant resources of therapeutic power, along with the accompanying wisdom of their usage as herbal remedies. The knowledge regarding this traditional medicine is accumulated in human society over time, shaped by the local culture and environment, and passed down from generation to generation. Even in this modern age, traditional medicine plays a significant role in primary health care and is found to be a promising source of novel drug candidates.

Herbal medicine indeed has a great impact on the present-day healthcare system worldwide, especially in developing countries. Herbal medicinal products meet the primary healthcare needs of about four billion people, representing 80% of the population of developing countries [3]. In contemporary society, herbal medicine is gaining more popularity, as synthetic drugs are expensive and have multiple side effects. In fact, the more secluded the place, the more its inhabitants are reliant on traditional herbal knowledge for their health care [1]. In spite of this, traditional herbal wisdom is steadily disappearing since it is transmitted verbally from one generation to the next. The lack of interest in herbal culture among younger generations or a lack of a sense of integrity between the two generations is the primary cause of the gradual disappearance of such traditional knowledge from society. On that account, traditional therapies need to be meticulously documented for future reference in order to preserve them permanently. For this reason, scientists regularly document folk herbal knowledge through diligent ethnobotanical studies from almost all the countries of the globe, including India. The Eastern Himalayan region of India has been studied by researchers for documentation of the rich herbal knowledge sustained in its diverse ethnic communities [4,5,6].

The Darjeeling district occupies a very small pocket of the eastern Himalayas, and it is situated in the northwestern part of West Bengal, a state in India. Due to favorable climatic conditions and physiographic diversity, the Darjeeling Himalayan region has a rich repository of plant species, including medicinal plants. Certain studies exploring ethnobotanical knowledge and associated phytoresources have been undertaken in this region [7,8,9]. None of the earlier works conducted in the district, except one, has employed quantitative tools in the statistical analysis of the ethnobotanical data for focused objectivity in this type of research [10]. For a rational bioprospecting program, scientists favor the ethnobotanical knowledge evaluated using appropriate statistical indices since ethno-guided leads or information give a higher hit rate than taxonomy-guided and randomly picked leads [11]. Therefore, our study aims to document and analyze knowledge about medicinal plants used by the tribal communities residing in the rural areas of the Darjeeling District, West Bengal, India, employing statistical indices such as fidelity level (FL%), informant consensus factor (F_ic_), and use value (UV) index through a semi-structured questionnaire.

Ethnomedicinal information plays a pivotal role in bioprospecting and exploring prospective drug molecules through the phytochemical and pharmacological screening of traditional herbal remedies. Moreover, statistically analyzed ethnomedicinal data in an ethnobotanical study can successfully guide phytochemists in choosing the most promising candidate among the documented medicinal plants for exploring their potential bioactive compounds [12,13]. In this context, the phytochemical study has opted to illustrate the phenolic and flavonoid profiles of the bark of *Betula alnoides* Buch.-Ham. ex D.Don, as this plant was found to be the most significant therapeutic agent in our current ethnobotanical investigation.

The general assumption is that the products derived from medicinal plants are inherently safer due to their natural origins, minimal processing, and time-tested uses [14]. Inadequate information on plant toxicity may result in the extended use of plants with intrinsic adverse effects on the user [15]. For the safe utilization of therapeutic plants and their extracts in human healthcare, it is vital to study the toxicity profile of medicinal plants. With respect to this, as part of a safety evaluation and CC_50_ value determination, an in vitro cytotoxicity study of the methanolic bark extract of *B. alnoides* has been performed on the L929 mouse fibroblast cell line.

In summary, the current study aims to document the wide array of phytotherapeutics traditionally used by the folk communities residing in rural pockets of the Darjeeling District and to elucidate the phytochemical and toxicity profiles of the methanolic bark extract of *B. alnoides*.

## 2. Results

### 2.1. Sociodemographic Profile of the Informants

Information on traditional phytotherapy practices was gathered from remote areas of the Darjeeling District by interviewing the selected local informants using a standard questionnaire. A total of 47 informants were interrogated during the survey, of which 11 were herbalists, locally called “baidya” in Nepali dialect. The herbalists were the key informants in this study, and the other 36 informants were knowledgeable in the therapeutic uses of local plant wealth. Sociodemographic profiles of the respondents demonstrate that the majority of them (25.5%) belong to the Rai community, followed by Limbu (21.3%), Thapa (19.1%), Khawas (10.6%), Lepcha and Chettri (6.4% each), Sharma and Tamang (4.3% each), and Sherpa (2.1%). The people who regularly practice ethnomedicine in their day-to-day lives were mostly males (65.9%), and few were females (34.1%). Among the respondents, 14 people were between 30 and 40 years old (29.8%), 7 people belonged to the age group of 41–50 years (14.9%), 15 people belonged to the age of 51–60 years (31.9%), 8 people ranged between 61 and 70 years old (17%), and 3 people fell into the age group of 71–80 years (6.4%). All the information, including the education, experience, and occupation of the respondents, is provided in Table 1. A list of the informants who were interviewed, along with the GPS coordinates of their residences and plant collection areas, is provided in Appendix A.

### 2.2. Ethnomedicinal Knowledge and Its Related Phytoresources

A large body of folk herbal knowledge, as well as associated phytoresources, were meticulously documented from the rural areas of the Darjeeling District after analyzing the recorded data statistically using certain ethnobotanical indices. Various aspects, like diversity, utility, and therapeutic potential of both the herbal knowledge and plant resources of the district, have been enumerated below.

### 2.3. Reported Medicinal Plant Species and Their Families

From the three subdivisions of the Darjeeling District, a sum of 115 species of medicinal plants were recorded, which spread over 65 families and 104 genera. Here, a total of 93 plant species belonged to dicots, 17 species were from monocots, 3 species were pteridophytes, and only 2 species were listed from gymnosperms (Table 2). The Asteraceae family had nine species, the highest number of species recorded, which covered 7.9% of the total number of species listed in the present study. The second largest families with respect to recorded species were Lamiaceae and Zingiberaceae, with six species and 5.3% of the total number of recorded species in each of these two families. The Fabaceae ranked the third largest family, with a total of four species, which represent 3.6% of the total species recorded. Each of the seven families with three species, accounting for 2.6% of total species, belonged to the families Amaryllidaceae, Apocynaceae, Lauraceae, Poaceae, Malvaceae, Myrtaceae, Piperaceae, Urticaceae, and Combretaceae. Each of the eleven families of Saxifragaceae, Solanaceae, Asparagaceae, Euphorbiaceae, Polypodiaceae, Phyllanthaceae, Plantaginaceae, Rosaceae, Polygonaceae, Apiaceae, and Menispermaceae represented two species, accounting for 1.7% of the total plant species recorded. The remaining 41 families had single-species representation, constituting 19% of the total noted species for each family (Figure 1).

The reported plant taxa are classified into four categories based on their habits, namely, herbs, shrubs, climbers, and trees. From the present study, the herbs contributed the highest number of species, 57, which shared 51.4% of the reported taxa. Trees represent the second largest group, with a total of 26 species and 23.4% of total species recorded, followed by the shrub group, with 17 species (15.3%), and climbers, with 11 species (9.9%) (Figure 2).

### 2.4. Plant Part(s) Used as Crude Drugs

Different parts of the recorded medicinal plants are used as crude drugs by the local people of the study area. The leaf was found to be the most used plant part in the current study, accounting for 31.2% of the ethnomedicinal formulations recorded, followed by underground parts like the root, rhizome, and bulb, which were used in 19.4% of the formulations. Fruits were used in 12.4% of the formulations, the bark and stem were used in 9.4% each, the whole plant was used in 8.8%, the flowers were used in 5.3%, and the seeds were used in the preparation of 4.1% of the total reported formulations (Figure 3).

### 2.5. Disease Categories

Based on the disease classification by the International Classification of Primary Care [16] of the World Health Organization (WHO), the documented 50 types of diseases have been assembled into 11 disease categories, which are (1) pregnancy, childbearing, and family planning, (2) digestive, (3) cardiovascular, (4) skin, (5) urological, (6) general and unspecified, (7) musculoskeletal, (8) respiratory, (9) endocrine/metabolic and nutritional, (10) blood, blood-forming organs, and immune mechanism, and (11) eyes. In the current study, it was noticed that the highest number of medicinal plant species, 23 species, were used in healing digestive diseases. For cardiovascular diseases, 18 species were employed, followed by 14 plant species administered for treating skin diseases, 12 plant species applied against general and unspecified diseases, and 8 species used for treating musculoskeletal problems. The lowest number of medicinal plant species, two species, was recorded for the category of pregnancy, childbearing, and family planning.

### 2.6. Procurement, Formulations, and Methods of Preparation of the Crude Drugs

Fresh plant materials sourced from wild and home gardens were utilized by the inhabitants of the studied area for preparing various herbal formulations. Among the 115 recorded species, 86 plant species were grown in the wild. This demonstrates the abundance of therapeutic plants in the natural habitat and also highlights the local people’s reliance on wild plant sources. In addition, crude herbal materials were procured by the local people from 29 species grown in their home gardens.

To treat 50 different kinds of diseases, a total of 122 formulations were administered. Among the recorded formulations, 101 formulations were monoherbal, and the remaining 21 formulations were polyherbal, which were prepared using more than one herbal species. Different parts of the 91 plant species are used for the preparation of 101 monoherbal formulations, and a total of 41 species are employed in preparing 21 polyherbal formulations. In the monoherbal group, nearly 10 species are employed in preparing more than one remedy administered for treating two or more ailments. Similarly, 10 species have been identified from the polyherbal group, which are used in the preparation of more than one formulation. There are 17 species commonly found in both mono- and polyherbal preparations (Table 3). When analyzing the various medicinal uses of the recorded 115 species, it was noticed that 28 species have multiple therapeutic uses. Each species has a minimum of two and a maximum of four types of uses for therapeutic purposes (Table 4). The larger part of the therapeutic formulations (81%) was constituted using fresh herbal ingredients. Some remedies were prepared entirely from dried plant materials (16.5%), while 2.5% of formulations were made of either air-dried or freshly harvested plant components, depending on their availability in the study area.

It was noticed that respondents in the study area used numerous methods of recipe preparation. Four different methods were employed for the preparation of remedies. The most common method of medicine preparation was juice (37.1%), followed by paste (35.3%), decoction (18.1%), and powder (9.5%) (Figure 4).

The recorded medicinal plant species were tabulated and presented alphabetically, followed by their family name, voucher number, vernacular name, diseases treated, plant part(s) used, mode of preparation, and method of administration in two separate tables in terms of monoherbal and polyherbal formulations (Table 5 and Table 6). 

### 2.7. Routes of Administration of Ethnomedicine and Doses

Around 75.6% of medications were administered orally. The root, bark, and leaf of certain plant species were boiled, and the decoction was consumed orally to heal diseases like diarrhea, body aches, fever, cough, throat pain, and others. Topical methods, such as massage, poultice, paste, and so on, were applied to treat various types of ailments, like bone fractures, skin problems, snakebites, cuts, etc. (Figure 5). 

The informants utilized their traditional system of measurement for measuring the amounts of drug components and doses of crude drugs with the help of teaspoons, cups, pinches, fingers, etc. This age-old technique of measurement is significantly supportive to traditional herbalists in measuring the amounts of plant resources needed for the preparation of medication and defining the precise dosages to be given to a patient for effective treatment [17]. However, dosages varied from disease to disease, as well as due to other factors, like the gender, age, and health condition of the patient.

### 2.8. Quantitative Analysis of the Ethnomedicinal Data 

Quantitative ethnobotanical indices offer organized and quantifiable information on the traditional uses of medicinal plants, which are crucial to ethnopharmacology research. Based on the frequency and range of the applications of plants in traditional medicine, these indices provide a structured method for measuring their significance in the health care system and utilitarian culture of a society. The quantitative indices offer statistically analyzed data that enhance the credibility of the documented information needed for drug discovery. In this context, the ethnomedicinal data of the present study were collected and analyzed utilizing suitable quantitative indices, such as the informant consensus factor (F_ic_), fidelity level (FL%), and use value (UV).

#### 2.8.1. Informant Consensus Factor (F_ic_)

F_ic_ is a highly favored quantitative tool used in ethnobotanical studies to indicate the plants that are utilized with higher to lower levels of accord among the informants to treat the diseases in a certain disease category [16]. The F_ic_ values of 11 recorded disease categories ranged between 0.56 and 0.87. The digestive diseases had the highest F_ic_ score of 0.87 among the eleven disease categories. It indicates that there is the highest degree of concurrence of digestive system diseases among the inhabitants of the study area. A total of 23 plant species are used for treating this diseases category. The smallest F_ic_ value was calculated in the case of endocrine/metabolic and nutritional diseases (0.56). Here, pregnancy, childbearing, and family planning gained a F_ic_ value of 0.83, followed by a F_ic_ value of 0.82 estimated for urological diseases. In these two categories of diseases, a high level of agreement among the informants is evident regarding the uses of the cited species of medicinal plants. The categories of respiratory and eye diseases had equal F_ic_ values of 0.78, followed by the category of general and unspecified diseases (0.72), etc. (Table 7).

#### 2.8.2. Fidelity Level (FL%)

The FL% value of the 91 noted plant species ranged between 29% and 97%. The highest FL% value (97%) was documented for one plant species, *Betula alnoides* Buch.-Ham. ex D.Don. The bark paste of this plant is topically applied for treating snakebites, inflammation, cuts, and wounds. The lowest value for FL% was 29%, which was calculated for the plant *Mallotus philippensis* (Lam.) Müll. Arg. The bark decoction of this medicinal plant is taken orally for treating piles (Table 6).

#### 2.8.3. Use Value (UV)

The UV index values for the recorded 91 species of medicinal plants varied from 0.02 to 0.97. *Hellenia speciosa* (J.Koenig) S.R. Dutta exhibited a use value of 0.97, which is the highest among the studied 91 species. Traditionally, the stem juice of this plant is taken orally to cure urinary troubles and jaundice. On the other hand, *Scadoxus multiflorus* (Martyn) Raf. scored the lowest use value, which was 0.02. The bulb paste of this plant is applied topically for treating mumps (Table 6).

### 2.9. Quantitative Estimation of Phenolic, Flavonoid, and Tannin Contents of B. alnoides Bark

Phenolics are widely acclaimed for their broad-spectrum pharmacological properties. The total phenolic content of the methanolic extract of *B. alnoides* bark was estimated to be 87.8 ± 2.5 mg gallic acid equivalent (GAE)/g dry tissue. This result was obtained from a calibration curve (y = 0.0199x − 0.064, R^2^ = 0.994) of gallic acid. The total flavonoid and tannin contents of the bark extract were estimated to be 30.1 ± 1.5 mg catechin equivalent (CE)/g dry tissue (y = 0.017x + 0.012, R^2^ = 0.9973) and 10.37 ± 0.51 mg tannic acid equivalent (TAE)/g dry tissue (y = 0.403x + 0.0419, R^2^ = 0.992), respectively.

### 2.10. HPLC Analysis of the Bark Extract

In this experiment, a total of six phenolic compounds have been identified from the *B. alnoides* bark, and the quantification of each of those six compounds has been made using the HPLC technique. Among the six identified compounds, the most abundant compound was chlorogenic acid, with a concentration of 117.5 mg/g of extract, followed by sinapic acid at 64.7 mg/g of extract. Caffeic acid was also present but in a lower amount of 5.8 mg/g of extract. Additionally, coumarin was present at a concentration of 1.5 mg/g of extract, p-coumaric acid at a concentration of 0.4 mg/g of extract, and gallic acid at a very minimal concentration of 0.1 mg/g of extract (Table 8; Figure 6).

### 2.11. In Vitro Cytotoxicity Assay

The methanolic bark extracts of *B. alnoides* were tested at concentrations ranging from 50 to 300 mg/L employing the MTT assay, and the impact on the cell line was evaluated after 24 h of cell exposure to the extracts. The results demonstrated a concentration-dependent increase in cell mortality, with percentages of 10.8 ± 0.45% at 50 mg/L, 21.08 ± 0.51% at 100 mg/L, 31.03 ± 0.49% at 150 mg/L, 40.21 ± 0.66% at 200 mg/L, 45.11 ± 0.74% at 250 mg/L, and 54.17 ± 0.57% at 300 mg/L (Appendix A). Toxic effects were observed at all concentrations higher than 100 mg/L, as indicated by a change in cell morphology from its typical spindle shape to an oval shape (Figure 7). A statistical analysis revealed a significant positive correlation (*p* < 0.05) between extract concentration and cell mortality, with a clear revelation of increasing cell mortality percentage with respect to enhancing the extract concentration. A non-linear regression analysis estimated the CC_50_ value of the methanolic bark extract to be 270.07 ± 1.32 mg/L, highlighting the potent effects of the extract on cell viability.

## 3. Discussion

The interaction between humans and plants is an inevitable aspect of the biological world. For centuries, humans have relied on plants as primary sources of food, shelter, and therapeutic medicaments. The age-old practices of utilizing medicinal plants and their diverse applications in treating various health problems have been deeply ingrained within indigenous cultures of mankind worldwide. The current study documented the therapeutic utilities of 115 species of vascular plants, which spread across 65 families of angiosperms, gymnosperms, and pteridophytes from the rural areas of the Darjeeling District of West Bengal. A total of 101 monoherbal formulations and 21 polyherbal formulations were recorded, and both types of formulations were prepared by the local people using plant ingredients from 115 species. This amply illustrates the diversity of folk herbal knowledge and the abundance of ethnomedicinal plants in the current study area. A thorough analysis of various therapeutic uses of all the recorded 115 medicinal species revealed that a total of 28 species have multiple uses, ranging from two to four therapeutic purposes. These 28 species of medicinal plants constitute an indispensable phytoresource in the locality, which needs a sustainable management strategy for its restoration and judicious utilization.

From our ethnomedicinal survey, it became evident that Asteraceae has the highest representation, with nine species utilized by ethnic groups in the studied area. This aligns harmoniously with many prior ethnomedicinal studies, which consistently highlight the prevalence of Asteraceae species in traditional medicinal practices [32,33,34]. *Bidens pilosa*, from the family Asteraceae has historically served as a remedy for liver ailments and hypertension in Taiwanese folk medicine [35]. However, within our surveyed region, the plant finds application in the treatment of skin infections through the topical application of its leaf paste. In the Nigerian traditional medicine of Africa, *Ageratina adenophora* is renowned for its efficacy in managing fever, diabetes, and inflammation [36]. Conversely, in our study, herbal medicine practitioners advocate for the oral administration of its leaf juice to cure piles. These poly-pharmacological properties of the species belonging to the Asteraceae family have been attributed to their medicinally potent phytochemical components, including essential oils, lignans, saponins, polyphenolics, sterols, and some polysaccharides [37]. The multifaceted bioactive compounds, pharmacological properties, and abundance may account for the extensive utilization of the Asteraceae family in medicinal practices in the area. The second highest number of species documented in the present study is from the families Zingiberaceae and Lamiaceae, each representing six species. The abundance in the locality and therapeutic efficacy made the recorded species of these two families a suitable choice of use by the local people as crude drugs in their traditional medicine. The therapeutic performances of such plant species of Zingiberaceae and Lamiaceae may be explained by the scientific evidence reported earlier on their bioactivity-related phytochemical profiles, which include medicinally important chemical groups such as phenolics, flavonoids, tannins, saponins, triterpenoids, essential oils, and alkaloids [38,39,40]. 

This study also clarifies that local people of the current investigated region mostly use herbs (51.4%), among other life forms of the recorded plants, to treat a range of illnesses. The possible explanation for employing herbs in maximum numbers is that herbs are grown exuberantly in the area and are easily accessible [41]. Herbs are also easier to harvest than trees and other woody species since they are smaller and have a shorter height. It is a fact that individuals are more likely to explore plants for food and pharmaceutical items that are procured effortlessly and are available abundantly and in nearly all seasons of the year [42]. These factors have led to a significant proportion of herbaceous plants being utilized as therapeutic agents in nearly all traditional medical systems, including ethnomedicine worldwide.

The ethnomedicinal practices mostly rely on plants that grow in the wild. All ethnic tribes continue to use native wild plants extensively, and traditional medicinal practices are mostly based on these plants [43,44]. In the present study, a major portion of the medicinal ingredients (74.7%) were acquired by local people from the plant communities grown in the wild.

The present study concludes that respondents mostly favor the usage of leaves (31.2%), followed by underground parts (19.4%), to prepare their herbal medications. Leaves are mostly preferred, as they are available almost year-round and more easily harvestable than other parts of the plant, such as flowers, fruit, seeds, bark, and sometimes, roots. It has also been demonstrated in a number of previous studies all around the globe that leaves are the ingredient most frequently used to prepare traditional remedies for the treatment of different illnesses [45,46,47]. Ethnic herbalists mostly rely on leaves because the foliage of medicinal plants synthesizes and often stores a variety of therapeutically beneficial phytochemicals, which include vitamins, minerals, alkaloids, phenolics, tannins, terpenoids, and others [48].

The present study shows that among 50 types of diseases recorded, the most common type is digestive disease. Many digestive diseases, such as diarrhea, jaundice, liver problems, flatulence, and stomach discomfort, have been seen as prevalent in this area, likely due to variations in lifestyle, lack of proper hygiene, unsafe drinking water, types of food, including leafy green items, consumed, and the compatibility of these foods with individual digestive systems. In various previous studies, digestive diseases like flatulence, indigestion, dyspepsia, and constipation have been documented as predominant health problems in India and all around the globe [49,50,51]. 

The informants in this district use various methods of preparation of remedies to treat diseases. Based on the aforementioned results, it has been concluded that the most common technique of preparing remedies is juice (37.1%), in comparison to other methods, such as pastes (35.3%), decoctions (18.1%), and powders (9.5%). Due to its effectiveness and ease of preparation, juice may be favored in most of the folk medicine system [52]. Additionally, juice may have a greater concentration of active ingredients compared to other preparations of remedies like decoctions, powders, and sometimes, pastes [53]. This is because preparations other than juice are prepared through heat-aided processes, such as boiling, drying, etc., that can degrade, to some extent, many of the chemical compounds present in the crude drugs. Thus, juice retains the exact amounts of the beneficial chemical compounds that make it more efficacious. The application of juice as a remedy in a significant or higher percentage in comparison to other preparations has been reported in many previously published research publications on ethnobotany [54,55]. 

The efficacy potential of remedies is higher when they are prepared by employing freshly collected plant ingredients than when they are prepared from dried crude drugs. Bioactive compounds in crude drugs can undergo thermal degradation when exposed to high temperatures during boiling, or the photosensitive compounds may undergo photolytic degradation once the crude plant materials are exposed to sunlight during the drying process. Furthermore, a lack of care during drying may lead to microbial growth, resulting in spoilage of the crude drug and denaturation of its certain chemical ingredients. Therefore, fresh materials are preferred for preparing remedies to ensure maximum potency and safety. The fact that materials in both fresh and dried forms are utilized in the preparation of ethnomedicine increases the likelihood that informants will have year-round access to the ingredients used in herbal formulations. 

In the present study, it was noted that the oral route of remedy administration is more prevalent (75.6%) than other routes, like topical (23.1%) and nasal inhalation (1.3%). Similar to this study, numerous previous investigations have highlighted that in the majority of cases, folk medicaments are administered through the oral route [43,56,57]. Oral administration is usually practiced, as it makes it convenient to use and consume the medicine, and it is more effective. Along with oral administration, topical application remains an important method of herbal drug administration to cure ailments such as wounds, skin diseases, and body pain. Topical use, specifically poultice, enhances blood circulation in the affected areas and provides a protective layer, shielding infected wounds or sores from further microbial infection. Additionally, the herbal ingredients in the poultice contain antiseptic essential oils, phenolics, tannins, and other bioactive chemicals that penetrate the dermal tissues, aiding in the fight against microbial infection and reducing inflammation. This ultimately promotes wound healing [41]. 

The informant consensus factor (F_ic_) is assigned to measure the consensus of informants regarding the uses of the plants in a particular disease category. In the present study, digestive diseases had the highest F_ic_ value (0.87), which adheres to the similar trend of the highest F_ic_ value being estimated to be 0.95 for the digestive disease category in another previous study [58]. Other categories of diseases with higher F_ic_ values in this region are pregnancy, childbearing, and family planning (0.83), followed by urological diseases (0.82). Similar results with high F_ic_ have been encountered for these disease categories in some earlier reports [5,59]. The lowest F_ic_ value (0.56) was calculated in the case of the endocrine/metabolic and nutritional disease category. The low F_ic_ suggests lesser consensus among the informants regarding the use of medicinal plant taxa in this disease category. 

The analysis of F_ic_ data reveals that digestive system disorders are the most prevalent disease category in the region, with 23 plant species used to treat 11 types of complications associated with the digestive system, including stomach pain, gastrointestinal infection, diarrhea, flatulence, and gas problems. The use of these plants for treating digestive disorders in this region can be attributed to their rich phytochemical compositions and biological activities, which are validated by the existing literature.

*Acorus calamus*, one of the 23 plant species used to treat digestive disorders, has shown promising potential in treating gastrointestinal disorders due to its high contents of phenolic compounds like chrysin and galanginin, plant extracts that have been evaluated by earlier works. The pharmacological studies indicate that both methanol and water extracts (15 mg/kg) significantly reduce diarrhea in mice by inhibiting Na+ K+ ATPase activity. Additionally, the ethanol extract (500 mg/kg) has demonstrated anti-ulcer activity in mice by inhibiting acid secretion in the stomach [60,61]. Similarly, drug parts of the plants *Cymbopogon citratus* and *Kaempferia galanga* are found to be very common ingredients in the remedies used by the local people of the surveyed areas for healing diarrhea, stomach pain, and flatulence. Research has confirmed that citral, a monoterpenoid from *Cymbopogon citratus*, and kaempferol, a flavonoid from *Kaempferia galanga*, possess potent anti-inflammatory, antidiarrheal, and antioxidant properties, which justifies the inclusion of these medicinal plants in the disease category of digestive system disorders [62,63]. Moreover, *Houttuynia cordata*, a notable therapeutic remedy included in the plant list of digestive disease groups used for jaundice, has been established as an effective hepatoprotective agent by checking its ability to prevent CCl4-induced liver damage in mice by restoring the damaged hepatocytes. The hepatoprotective activity of this plant may be related to the presence of various antioxidant molecules of the polyphenolic group, namely, quercitrin, quercetin, rutin, hyperin, isoquercitrin, β-myrcene, β-pinene, α-pinene, α-terpineol, and n-decanoic acid, identified through phytochemical analysis [64]. A critical review of their phytochemical composition and biological activities suggests that most of these ethnomedicinal species contain therapeutically effective phytochemicals, justifying their traditional uses in addressing various disorders associated with the digestive system

On the basis of the score of the use value index (UV), *Hellenia speciosa* (J.Koenig) S.R.Dutta has been identified as having the highest use value, 0.97, among 91 species, which highlights the frequent use of this species in multiple disease conditions, i.e., for urinary troubles and jaundice. The lowest score was calculated for *Scadoxus multiflorus* (Martyn) Raf., which was 0.02. Such a low UV score for this plant illustrates that it is used infrequently for the least number of diseases; in this study, it is used for curing only one health condition, mumps. Many ethnobotanical studies employing quantitative indices, including the use value index, documented the higher to lower range of UV scores elucidating the degrees of importance of medicinal plant species with regard to their therapeutic uses [43,65].

The fidelity level (FL%) is a useful quantitative tool to determine the most beneficial plant species used for treating a certain disease. A higher percentage of fidelity level (FL%) can show that the utilization of a plant for specific therapeutic purposes is preferred if respondents cited it regularly. Plants with high FL% values become the torchbearers, leading us on a journey of scientific discovery through phytochemical studies to identify the bioactive compounds that are accountable for their exceptional therapeutic properties, as well as to validate the traditional claims on the medicinal uses of those species [66]. Over the course of our fidelity level analysis, *Betula alnoides* Buch.-Ham. ex D.Don emerged as the ethnomedicinal plant with the highest FL% of 97% among the 91 studied species, which indicates its excellent therapeutic potentialities. Recognizing that it is an exceptional attribute, we have embarked on studies of the phytochemical properties and cytotoxicity profile of *Betula alnoides* bark.

Almost all the previous studies from the Darjeeling District of West Bengal have documented the ethnobotanical knowledge on the traditional therapeutic uses of plant resources, without quantitative analyses of the collected data using dedicated ethnobotanical indices [6,7,8]. Notably, only one ethnomedicinal work from the Teesta Valley of the Darjeeling District employed quantitative indices like the informant consensus factor, fidelity level, and importance value in the analysis of data on ethnomedicinal plants [10]. In contrast, the novelty of the present study is that it focuses on the documentation of medicinal plant resources from three subdivisions of the Darjeeling District after analyzing the recorded information utilizing the quantitative indices of use value index, alongside the fidelity level and informant consensus factor, to measure the utilitarian significance and elucidate the usage patterns of the phytotherapeutics.

The present comprehensive documentation of folk herbal knowledge from the Darjeeling District underscores the vital importance of preserving the traditional wisdom of the district and harnessing its therapeutic potential to develop modern medicine. This research reveals the district’s remarkable ethnobotanical plant richness, cataloging 115 plant species that are integral to the traditional healing practices. Among the documented 115 plant species, a significant number of 103 species have been studied to some extent for their bioactivities, as well as their phytochemical profiles. This indicates a great promise of the documented phytoresources for further scientific investigation toward the development of a varied range of natural products. Notable examples of these species include *Achyranthes aspera*, *Bergenia ciliata*, *Bidens pilosa*, *Berberis napulensis*, *Datura stramonium*, *Kaempferia galanga*, *Oroxylum indicum*, *Euphorbia hirta*, *Phyllanthus emblica,* and many more. For instance, *Achyranthes aspera* exhibits significant anti-inflammatory and antioxidant effects [67], while *Berberis napulensis* has been found to be effective in cardiovascular, metabolic, hepatic, and renal disorders due to the presence of the alkaloid berberine [68]. *Kaempferia galanga* shows both wound healing and anti-inflammatory activities [69], and *Oroxylum indicum is* noted for its broad range of bioactivities, such as antioxidant, anti-inflammatory, anticancer, and anti-ulcer bioactivities, and many more [70]. In addition to these, a number of pharmacologically well-studied medicinal plants have also been documented from the Darjeeling District, including *Moringa oleifera*, *Catharanthus roseus*, *Calotropis gigantea,* and others. *Moringa oleifera* is known for its antioxidant [71], anti-inflammatory [72], and antimicrobial properties [73], with its bioactive compounds widely utilized in various health products, such as energy drinks (e.g., Zija’s Moringa-based XM3) and supplements (e.g., Kuli Kuli’s Moringa-infused energy bars). *Catharanthus roseus* is a vital source of potent anticancer alkaloids, vinblastine, and vincristine used in a broad range of cancer diseases [74], while *Calotropis gigantea* is revered for its hepatoprotective and wound healing properties [75], with its extracts used in topical ointments for wound care [76]. However, twelve species were found unexplored for their phytochemical and pharmacological evaluation among the documented species in this study. These twelve plant species, like *Allium rhabdotum*, *Entada gigas*, *Hydrangea febrifuga*, *Hydrocotylehimalaica*, *Hedychium spicatum*, and *Pseudognalium affine*, provide exciting avenues for scientific exploration toward natural product development. Targeted phytochemical analyses, bioactive compound isolations, and pharmacological evaluations of these underexplored species could unlock their full array of therapeutic potential for the development of health-promoting products. 

There are several studies detailing the phytochemical and pharmacological aspects of many folk medicinal plants from the Darjeeling District in which *Betula alnoides* has not been considered [77,78,79]. Some studies have explored various phytochemical groups, including the terpenoids, phenolics, and flavonoids of *Betula utilis* and *B. pendula* [80,81]. Only one study focuses on the quantification and chemical structure elucidation of a triterpenoid compound named lupeol isolated from *B. alnoides* bark [82]. The present research, however, quantified a total of six phenolic compounds from *B. alnoides* bark through the HPLC technique. A quantitative phytochemical analysis highlights the bark of *B. alnoides* as a rich source of phenolics and flavonoids. The phenolic content of *B. alnoides* bark was estimated to be 87.8 ± 2.5 mg GAE/g of dry tissue, which is greater than the phenolic contents of the bark of many well-recognized medicinal plants, like *Acacia nilotica* (80.6 mg GAE/g dry tissue), *Acacia catechu* (78.1 mg GAE/g dry tissue), *Senna tora* (65.5 mg GAE/g dry tissue), and *Cassia fistula* (22.8 mg GAE/g dry tissue) [83]. Our study also estimated the total flavonoid content in the bark of *B. alnoides,* and it was 30.1 ± 1.5 mg CE/g of dry tissue. The total content of flavonoids in *B. alnoides* bark is found to be almost two times higher than the total flavonoid content, i.e., 15.1 mg CE/g of dry tissue of bark of *Terminalia arjuna*, which is an important medicinal plant [84]. The bark extract showed a substantial tannin content of 10.37 ± 0.51 mg TAE/g of dry weight. As we documented earlier, during our ethnomedicinal survey, the ethnic communities of the Darjeeling District traditionally use the bark of the *B. alnoides* plant as an antivenom, anti-inflammatory, and wound-healing agent. Through phytochemical estimation, we have found that the bark of this plant possesses significant amounts of phenolics, flavonoids, and tannins, which are known to manifest a wide range of pharmacological activities, including antivenom, wound healing, and anti-inflammatory properties. Many research groups [85,86,87] have scientifically established that plant-sourced phenolic, flavonoid, and tannin compounds possess strong antivenom activity [88,89]. Numerous scientific reports have highlighted the effectiveness of flavonoids as wound-healing agents due to their high antioxidant and antimicrobial capacities [90]. There has been substantial evidence supporting the wound-healing potential of flavonoid-rich fractions of plants such as *Tephrosia purpurea*, *Martyni aannua*, *Ononidis radix*, and *Eugenia pruniformis* [91,92,93,94]. Tannins are also reported to facilitate wound healing due to their astringent properties, promoting wound tissue contraction, reducing microbial growth, neutralizing free radicals, and enhancing the growth of new blood vessels and fibroblast cells [95]. Scientific evidence that showcases the significant anti-inflammatory potential of phenolics, flavonoids, and tannins is abundant [96,97,98,99]. Therefore, the bark drug of our investigated plant possesses a very good amount of phenolics, flavonoids, and tannins, which qualifies it as a promising candidate for the development of effective anti-inflammatory and wound-healing agents. 

Further, the HPLC study revealed the presence of six phenolic compounds. Among these, chlorogenic acid (117.5 mg/g dry tissue) and sinapic acid (64.7 mg/g dry tissue) have been identified in the highest quantities in the bark extract. The broad-spectrum therapeutic values of all six detected phenolic compounds are well-documented in the scientific literature (Table 8). Particularly, the presence of a substantial amount of chlorogenic acid in the bark extract of the studied plant justifies its antivenom and wound healing properties. Research findings provide evidence for the antivenom activity of chlorogenic acid through the inhibition of the neuro-myotoxic activity of the phospholipase A2 enzyme [19]. Research on the topical use of chlorogenic acid on wounds in Wistar rats has shown that it can enhance wound healing. Additionally, the antioxidant properties of this compound contribute to the healing effect too [20]. The bark of *B. alnoides* is rich in sinapic acid, which may contribute to the wound-healing activity of the plant. Gels containing sinapic acid promote the healing of wounds in Wistar rats through re-epithelialization and reducing oxidative stress [22]. The presence of six well-established anti-inflammatory phenolic compounds [23,25,28,30] in the bark of *B. alnoides* validates its use by the local tribes of the surveyed area for treating various inflammatory conditions in humans. Therefore, it can be concluded that the high contents of phenolic and flavonoid compounds and the presence of those six phenolic compounds in the bark of *B. alnoides* contribute to its therapeutic potential. 

Like phytochemical studies, experimental works on the bioactivity of *B. alnoides* are very scanty. Only a few studies were conducted on certain bioactivities of the bark and stem of this plant. In one study, the bark demonstrated significant antioxidant, antimicrobial, and anti-inflammatory effects, which are attributed to its high polyphenolic content, including flavonoids and phenolics. In the same study, the bark shows α-glucosidase inhibition, suggesting its antidiabetic potential [100]. The stem of *B. alnoides* exhibited the anti-HIV-1 integrase and anti-inflammatory activities in another work, highlighting its promise for antiviral and inflammatory treatments [101]. However, to fully harness the plant’s therapeutic potential, future studies are to be carried out to investigate its varied types of bioactivities more comprehensively.

The earlier reports on traditional uses of other species of *Betula* have documented that leaves of *Betula utilis* are used for curing urinary tract infections [102], and *B. pumila* fruits are administered to treat respiratory tract diseases [103]. However, there is no such report regarding the medicinal uses of the leaf and fruit parts of *B. alnoides* in folk medicine. Given the medicinal importance of the leaf and fruit parts of other *Betula* species, there is potential for the leaf and fruit parts of *B. alnoides* to be explored as crude drugs by ethnic communities of the present study area, as well as other regions of the globe. Previous phytochemical studies on leaf parts of different *Betula* species highlight their rich phenolic content. For instance, the leaves of *Betula pendula* contain significant amounts of total phenolics (10.8 mg GAE/g dry tissue) and six phenolic compounds, namely, catechin, p-coumaric acid, myricetin, quercetin, and naringenin [104]. Similarly, *Betula aetnensis* leaf extracts are also abundant in phenolics and flavonoids [105]. Despite the established phytochemical richness in the leaves of various *Betula* species, studies have to be conducted on the leaf and fruit parts of *Betula alnoides*. Phytochemical exploration of these two parts of *B. alnoides* could disclose the hidden treasure of phenolics, flavonoids, and other polyphenolics, thereby expanding our understanding of the phytochemistry and therapeutic applications of this medicinal plant.

Despite the extensive ethnomedicinal claims and the presence of therapeutically significant phytochemicals, it is crucial to conduct systematic toxicity studies of this medicinal plant to predict potential toxicity risks and offer scientific evidence for ensuring the consumption of this drug in humans, as well as other animals. Our study is the first report on the cytotoxicity effects of *B. alnoides* bark against the L929 mouse fibroblast cell line, as no previous research has been performed to understand the cytotoxic effects of this plant. It was observed that the percentage of cell viability decreased with the increase in concentrations of the plant extract. When comparing the CC_50_ value of the methanolic bark extract of *B. alnoides* (270.07 ± 1.32 mg/L) to other medicinal plant extracts, it becomes clear that *B. alnoides* has moderate cytotoxicity. For instance, the CC_50_ value of the methanolic leaf extract of *Azadirachta indica* (neem) is reported to be approximately 200 mg/L, indicating a higher toxicity compared to *B. alnoides* [106]. Conversely, the methanolic extract of *Moringa oleifera* has a CC_50_ value of around 400 mg/L, suggesting it is less toxic than the cytotoxicity of *B. alnoides* bark [107]. These comparisons emphasize the relatively moderate cytotoxic nature of the *B. alnoides* bark extract. Therefore, the application of *B. alnoides* bark extract in a therapeutic context should be considered carefully, particularly at higher concentrations, to avoid potential cytotoxic effects. Further acute and long-term chronic toxicological studies are necessary to explore the underlying mechanisms of the toxicity and to ascertain the safe dosage range for its potential medical applications.

## 4. Materials and Methods

### 4.1. Study Area

The current study was conducted in three subdivisions of the Darjeeling District, namely, Kurseong, Darjeeling Sadar, and Mirik. In general, Darjeeling Himalaya falls under subtropical per humid climates, with daily mean temperature varying from 17 °C to 8 °C. The region receives 3372.75 mm of rainfall on an annual average [108]. This region spans between 26°57′16″ N and 88°25′2″ E and covers an area of 2090 km^2^. Numerous ethnic communities, including Tamang, Chettri, Lepcha, Rai, Limboo, and others, reside in this region. The Darjeeling District has 21.52% Scheduled Tribes and 17.18% Scheduled Castes [109].

### 4.2. Data Collection

The ethnomedicinal data were collected by surveying 11 villages under three subdivisions of the Darjeeling District using standard procedures by frequent field trips made in different seasons of the year during June 2022 and October 2023 [110,111]. The surveyed villages are Rolak, Lanku, Mungpoo, Rangaroon Tea Garden, Shelpu, Sittong, Phuguri, 6th Mile, Samripan, Tarzam, and Rungli (Figure 8). A total of 47 individuals (31 males and 16 females) were interviewed using a free listing, focus group discussion, and semi-structured questionnaire after communicating the goal of this study and its conclusion clearly to the interviewees in the local Nepali language using the purposive sampling method [112]. Free prior informed consent (FPIC) was taken from each of the informants before the interview in written form. The plants’ native names, plant parts used, preparation of the crude remedy, administration of the crude medication, ailments healed, and so forth were noted. The permanent addresses and locations of the residences of each participant were recorded using a global positioning system (GPS). The images of the plants and sociodemographic data were also recorded.

### 4.3. Collection of Plant Specimen and Herbarium Preparation

Herbarium specimens were prepared from the collected plant samples of the recorded species using the standard protocol [113]. The herbarium specimens have been stored at the departmental herbarium (Department of Botany, Visva-Bharati, Santiniketan, India) for future reference.

### 4.4. Identification and Nomenclature of the Plant Species

The plant species were identified with the help of different flora of the Darjeeling District and its adjoining regions [114,115,116]. Finally, species identification was confirmed by the expert taxonomists, Prof. Monoranjan Chowdhury from the Taxonomy of Angiosperms and Biosystematics Laboratory, Department of Botany, University of North Bengal, Darjeeling, West Bengal, and Prof. Chowdhury Habibur Rahaman from Ethnopharmacology Laboratory, Department of Botany, Visva-Bharati, Santiniketan, West Bengal, India. The nomenclature of the collected species has been updated using websites like Plants of the World Online (https://powo.science.kew.org/) and the World Flora Online (https://www.worldfloraonline.org/).

### 4.5. Data Analysis

#### 4.5.1. Qualitative Data Analysis

The sociodemographic data of the informants were analyzed and documented in a table. All the recorded medicinal plant species were analyzed and grouped into various taxonomical categories. The recorded information on ethnomedicinal knowledge was tabulated and enumerated in tabular form along with the native name of the plant, updated scientific name, family, voucher specimen number, plant components utilized, diseases treated, technique of preparation of medicines, and the medicines’ administration with doses.

#### 4.5.2. Quantitative Data Analysis

The following quantitative tools were employed.

##### Informant Consensus Factor (F_ic_)

F_ic_ is a quantitative tool commonly applied in ethnobotany data analysis. It was first proposed by [117] and derived from the respondent agreement ratio equation, which was introduced by Heinrich et al. (1998) [118]. The F_ic_ value ranges between 0 and 1. The F_ic_ value assesses the maximum likely therapeutic plant species utilized by the informants for a particular disease category. A high F_ic_ value indicates greater consent among the respondents about the plant species used for a group of diseases. The following formula was used to obtain the F_ic_ value:F_ic_ = N_ur_ − N_t_/N_ur_ – 1(1)

Here, N_ur_ denotes the number of use reports in each group, and N_t_ denotes the number of taxa (species) recorded.

##### Fidelity Level (%)

The fidelity level (FL) is the percentage of respondents who cite the use(s) of a specific plant species to cure a specific disease(s). It is used to assess the percentage of informants who assert the usage of a particular plant species for healing a specific disease, and it was calculated using the following equation:FL(%) = N_p_/N × 100 (2)
where Np represents the number of interviewees who assert to use of a specific species to cure a particular ailment, and N represents the number of interviewees who use the plant as a medicine to cure any given disease. The highest FL% indicates that the plant species is applied recurrently and widely by the respondents to treat a specific health condition [119].

##### Use Value Index (UVs)

The use value index (UV) has been employed to determine the most recurrently utilized plant species [120]. The formula for determining the use value index for each species is UV=∑U/n.

Here, UV stands for the use value of species. U is the sum of the total number of use citations by all informants for a given species, and n is the total number of interviewees. A higher UV value designates the most utilized plant species and a lower UV value indicates the least utilized plant species.

### 4.6. Collection of Bark and Identification of the Plant

The fresh stem bark of *Betula alnoides* was collected on 22 October 2022 from an open forest near Lanku Valley of the Darjeeling District, West Bengal (23°31′56.4″ N and 87°30′36.8″ E), consulting the guidelines recommended by the National Medicinal Plants Board, New Delhi, India [121]. The identification of this collected species has been confirmed by consulting with different flora of Darjeeling and adjoining areas of West Bengal [114,116] and the expert taxonomist. The nomenclature of this plant has been made up-to-date following the standard website, such as Plants of the World Online (https://powo.science.kew.org/). The herbarium specimen has been prepared [110] and kept in the Herbarium of the Department of Botany, Visva-Bharati, Santiniketan, India, for future considerations [Voucher specimen number: INDIA, West Bengal, Darjeeling district, Lanku Valley forest, 15.11.2022, Y Subba 71 (VBYS 71)].

### 4.7. Preparation of Plant Extract

The collected bark of *B. alnoides* was washed thoroughly and then sliced into small thin pieces, shade-dried, and ground into fine powder. The plant powder was stored in an airtight vessel at 4 °C for future use. The bark powder of 10 g was extracted with 100 mL of methanol in a 250 mL conical flask keeping in a mechanical shaker at 28 ± 2 °C for 36 h. For a single extraction, the whole process was repeated three times. The resulting pull-out was mixed and filtered with Whatman’s No.1 filter paper. The filtrate was subjected to evaporation at room temperature (28 ± 2 °C). The ultimate yield was stored at 4 °C and dissolved in dimethyl sulfoxide (DMSO) to make the stock before use.

### 4.8. Quantitative Estimation of Phenolic, Flavonoid and Tannin Contents

The total phenolic content of the methanolic bark extract of *B. alnoides* was determined using the Folin-Ciocaletu method [122], while the total flavonoid content was estimated through the aluminum chloride colorimetric method [122]. The method of Afify et al. (2012) was employed to estimate the total tannin content [123].

### 4.9. HPLC Analysis of the Bark Extract

An HPLC device (Agilent 1260 Infinity II, Santa Clara, California, USA) equipped with OpenLab 2.7 data processing software (V2.7.) was used for the phytochemical analysis of the methanolic bark extract of *B. alnoides*. A reversed-phase column, Luna C18 (25 cm in length, 4.6 mm in inner diameter, and 5 μm in thickness) (Phenomenex, USA), was utilized for compound separation. The stock solutions (1 mg/mL) were prepared separately both for the standard and bark extract. The 1 mg of bark extract was combined with 0.5 mL of HPLC grade methanol and continuously sonicated for 10 min to prepare the stock solution. The overall volume of the mixture was made to 1 mL by adding the HPLC mobile phase solvent (which is a mixture of 1% aqueous acetic acid and acetonitrile in a 9:1 ratio). About 20 μL of the bark extract was introduced at a rate of 0.7 mL/min, and the temperature of the column was maintained at 28 °C during the analysis. For gradient elution, a variable percentage of solvent B (acetonitrile) to solvent A (1% aqueous acetic acid) was implemented. Within the initial 28 min, the ratio of gradient elution was altered linearly from 10% to 40% of solvent B, then from 40% to 60% of solvent B up to 39 min, and finally from 60% to 90% in 50 min. It required 55 min for the mobile phase’s composition to revert to its original state of solvent B/solvent A (10%:90%). It was then kept running for an additional 10 min before another sample was injected. The total time required to analyze each sample was approximately 65 min. Chromatograms of HPLC were recorded at three distinct wavelengths (272, 280, and 310 nm) using a photodiode UV detector. Based on the retention times and by matching with the applied standards, all of the phenolics and flavonoids were identified. A calibration curve was generated against various concentrations of the relevant standard samples in order to quantify the detected chemicals. Ten standards, including one flavonoid (kaempferol) and nine phenolic compounds (quercetin, chlorogenic acid, syringic acid, sinapic acid, catechin hydrate, caffeic acid, p-coumaric acid, coumarin, and gallic acid), were utilized for HPLC study.

### 4.10. In Vitro Cytotoxicity Assay

Using the 3-(4,5-dimethylthiazol-2-yl)-2,5-diphenyltetrazolium bromide (MTT) assay, the cytotoxicity test was conducted on the L929 mouse fibroblast cell line (sourced from the APT Research Foundation, Pune, India). Cells (10^4^ per well) were seeded into a 96-well tissue culture plate. The plate was kept in an incubator for 24 h at 37 °C with 5% CO_2_. The test sample solution was prepared by dissolving the methanolic bark extract of *B. alnoides* in Dulbecco’s Modified Eagle’s Medium (DMEM), with 10% fetal bovine serum (FBS). Wells containing cells with no extracts (DMSO only) were designated as a negative control. The prepared test sample solution was then added to each incubated well at different concentrations ranging from 50 to 300 mg/L and further incubated for 24 h at 37 °C with 5% CO_2_. After the incubation period, the plates were examined under an inverted microscope, to observe changes in cellular structure and photographs were taken. Then 10 µL of MTT reagent was added to each well of the plate and it was incubated for 4 h at 37 °C with 5% CO_2_. After the incubation period, the residual medium was removed, and 100 µL of solubilization buffer was added to each well. OD was taken at 630 nm using a 96-well plate reader. Each experiment was repeated three times. Cytotoxicity was analyzed using the following formula:
(3)$$\mathrm{Cytotoxicity}\;(\%)\;=\;(A_{o}-\frac{At}{Ao})\;\times \;100\;$$
where *A*_0_ is the negative control’s absorbance, and *A_t_* is the absorbance of the test sample solution.

CC_50_ (50% cytotoxicity concentration) value and the standard deviation (SD) were calculated using a non-linear regression curve obtained using Microsoft Excel, 2007.

## 5. Conclusions

The present study extensively documents a significant volume of indigenous herbal knowledge on medicinal plant resources which play a vital role in the health care of the local population in the Darjeeling District of West Bengal. The knowledge domain of ethnomedicine in the study region embodies about 115 plant species used in the forms of 101 monoherbal and 21 polyherbal formulations for therapies of 50 types of diseases. Our study amply elucidates the knowledge diversity of folk medicine in the district of Darjeeling and the richness of associated ethnomedicinal plant species with multiple therapeutic options.

One of the significant findings of this work is that among the 115 species, a total of 28 plant species with multiple medicinal uses have been found to be very promising for further chemical and bioactivity studies. Considering their multipurpose uses, such versatile species of medicinal plants should receive proper conservation and management attention to ensure their sustainable utilization. The present documentation will also be helpful to protect this valuable ethnomedicinal knowledge domain from its loss in the near future, as such herbal tradition remains in oral form, mostly among the senior members of the communities. The data presented in our work will enhance the ethnomedicinal database of this district, as well as the West Bengal state.

The quantitative indices offer statistically analyzed data on the medicinal uses of plants that amplify the credibility of the documented information crucial for bioprospecting and drug discovery. Analysis of the ethnobotanical data employing three quantitative indices measured the importance of recorded species in the herbal culture of the Darjeeling District, highlighting a good number of ethnomedicinal species with greater quantitative index values as the most significant and indispensable plant resources, for example, *Betula alnoides* (FL = 97%), *Hellenia speciosa* (UV = 0.92), and many others. Thus, the present study provides a big list of statistically analyzed important medicinal plants that show immense promise of being exploited for natural product discovery in the future.

A quantitative phytochemical analysis revealed a significant presence of phenolics and flavonoids in the bark of *B. alnoides*, highlighting its potential pharmacological benefits. The HPLC study further identified the bark of this plant as a noble source of chlorogenic acid and sinapic acid, as these two compounds were found in the greatest quantities among all six phenolic compounds identified. These phenolic acids, known for their broad pharmacological activities, including antioxidant, anti-inflammatory, wound healing, and antivenom properties, make the rationale to some extent regarding the traditional medicinal use of bark part of the investigated plant in snakebites, inflammation, cuts, and wounds. Furthermore, the MTT cytotoxicity study on the L929 mouse fibroblast cell line suggests that the bark extract of *B. alnoides* is likely to be moderately toxic, with a CC_50_ value of 270.07 ± 1.32 mg/L, and at higher concentrations, it may show cytotoxic effects. Future research efforts should focus on fractionating this bark extract to identify and isolate more bioactive compounds and conducting detailed toxicity studies to determine the safe dosage for potential applications. Utilizing advanced techniques such as molecular docking and in vivo studies will be crucial for fully elucidating the therapeutic potential of this highly potent plant.

There is no report on the uses of the leaf and fruit of *B. alnoides* for the healing of diseases. The tall height of mature trees (up to 30 m) of *B. alnoides* makes leaf and fruit collection very challenging, whereas bark is easier to harvest. While smaller trees and saplings of the plant offer access to leaves and fruits, the stem bark remains easily accessible; thus, it is the preferred plant part used for the preparation of traditional medicine. The herbalists and other users in the area ensure the sustainable use of the phytoresources, and while collecting the bark, they avoid ring-barking, which is a method that is detrimental to the plants, and instead collect small portions of the bark using a stripping method. So, sustainable collection practices are usually followed by the local inhabitants to protect the plant communities particularly the tree community in the area from the life-threatening harm caused by indiscriminate harvesting practices. During field surveys, a very scanty population of *B. alnoides* was noticed in some discrete patches, with few tree individuals in the area. The population decline of this plant in the area is mainly because of the cutting of the trees for timber. Therefore, conservation efforts should prioritize protecting the tree, given its multipurpose use. Proper conservation strategies should be undertaken to ensure the protection of this plant species from the rapid declination of its population.

## Figures and Tables

**Figure 1 plants-13-03505-f001:**
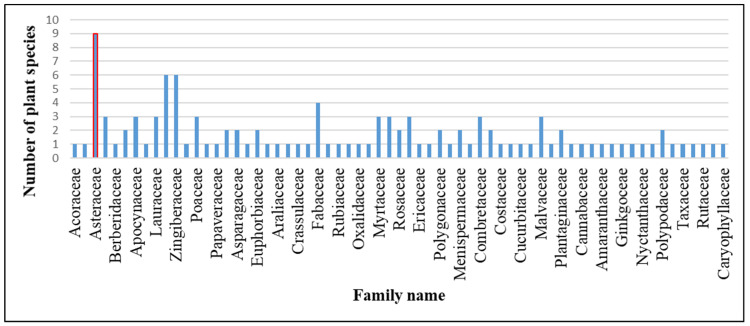
Number of plant species in respective families.

**Figure 2 plants-13-03505-f002:**
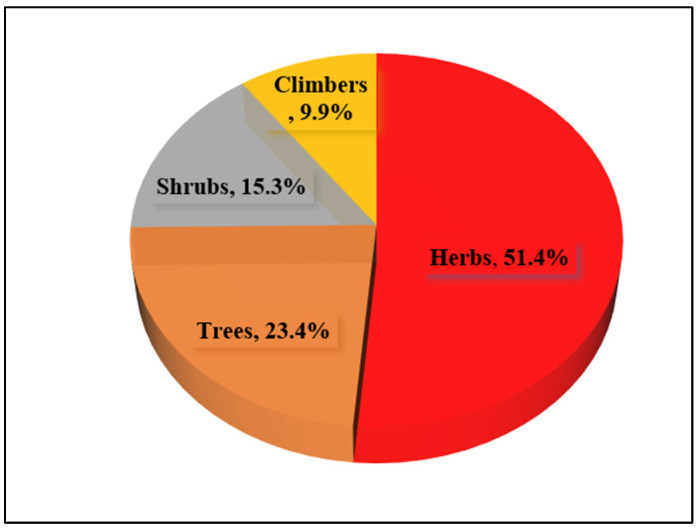
Habits of medicinal plants in the study area.

**Figure 3 plants-13-03505-f003:**
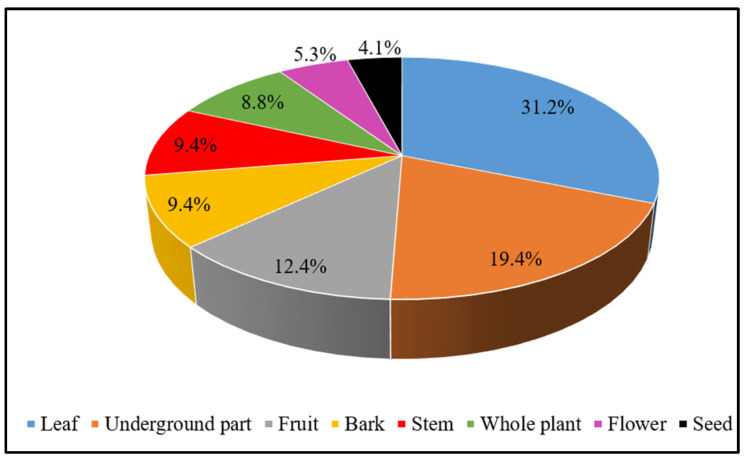
Plants parts used in herbal preparations.

**Figure 4 plants-13-03505-f004:**
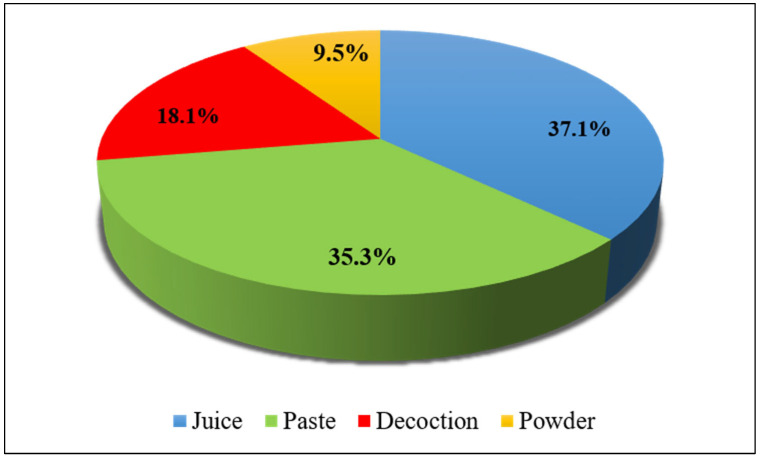
Methods of preparation of ethnomedicine.

**Figure 5 plants-13-03505-f005:**
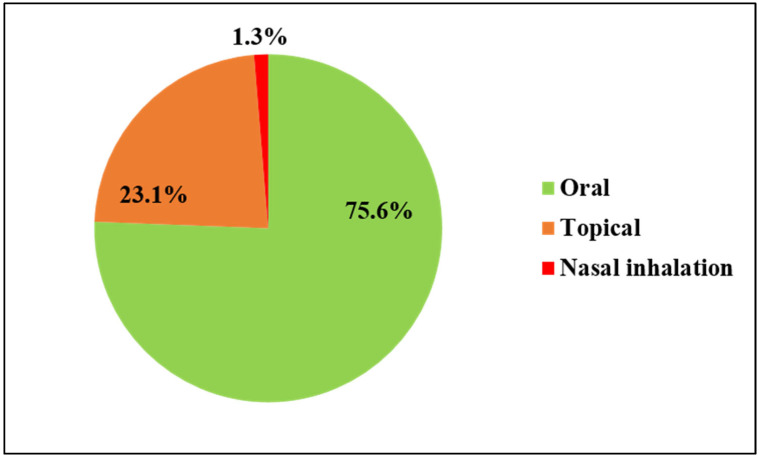
Route of administration of ethnomedicine.

**Figure 6 plants-13-03505-f006:**
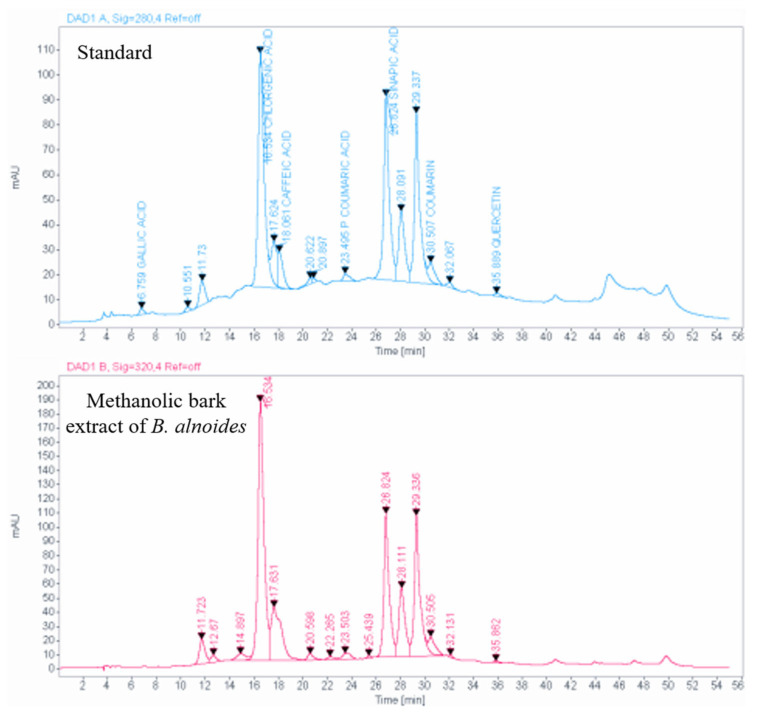
HPLC chromatogram obtained from methanolic bark extract of *B. alnoides*.

**Figure 7 plants-13-03505-f007:**
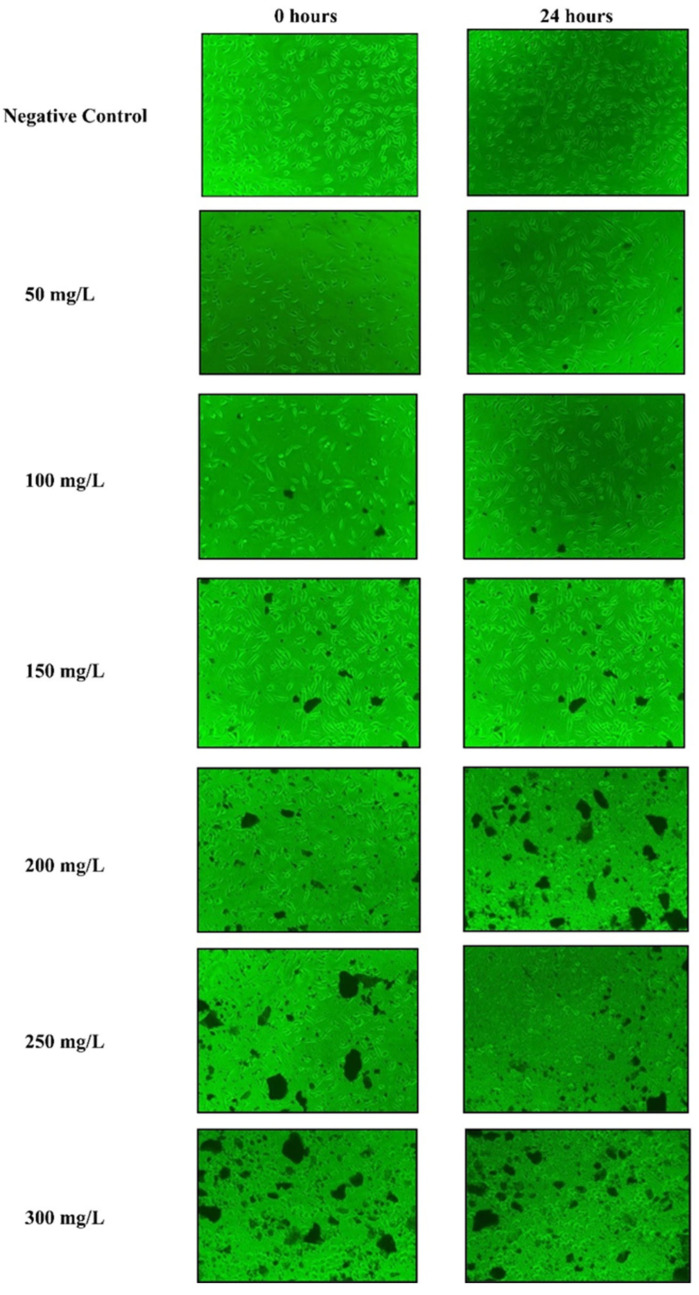
Micrographs showing the morphological changes in the L929 mouse fibroblast cells before and after 24 h exposure to different concentrations (50–300 mg/L) of *B. alnoides* bark extract, along with negative control.

**Figure 8 plants-13-03505-f008:**
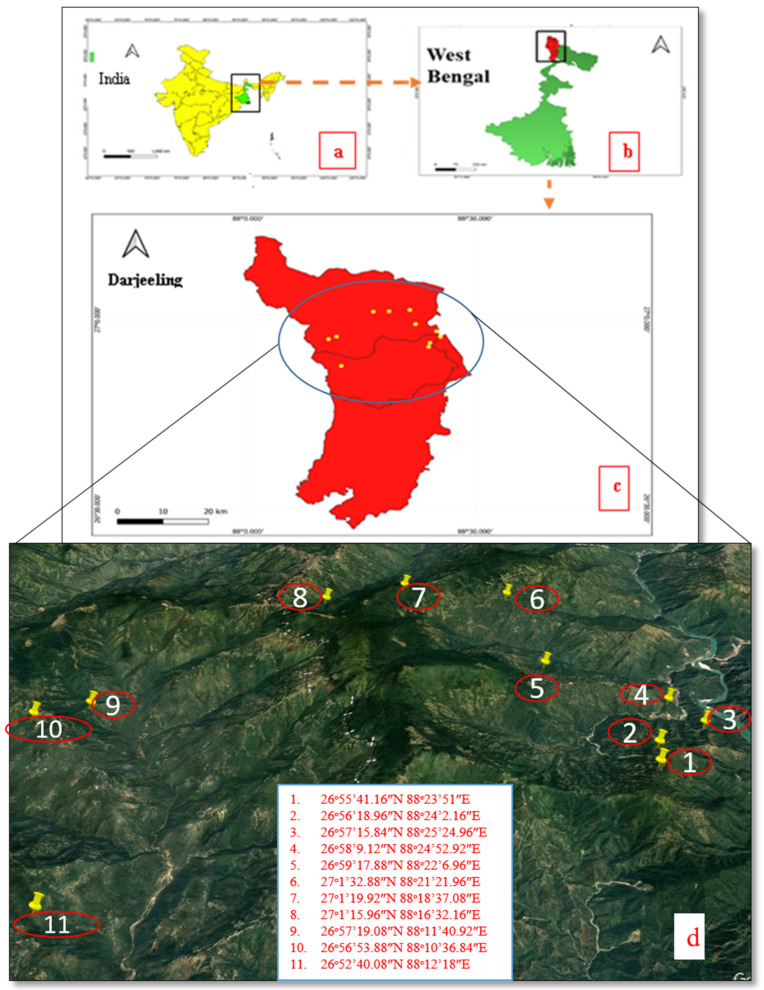
Map (**a**–**d**). (**a**) Map showing the geographical location of West Bengal in India. (**b**) Geographical location of Darjeeling District in West Bengal. (**c**) The map of surveyed area. (**d**) Map depicting the various villages visited during the present study in Darjeeling District, West Bengal. (The map is created using Qgis software 3.10. and Google Earth Pro 7.3).

**Table 1 plants-13-03505-t001:** Summary of the sociodemographic profiles of the informants (n = 47).

Variables	Categories	Numbers	Percentage (%)
Gender	Male	31	65.9
	Female	16	34.1
Ethnic community	Thapa	09	19.1
	Rai	12	25.5
	Lepcha	03	6.4
	Limbu	10	21.3
	Sharma	02	4.3
	Tamang	02	4.3
	Khawas	05	10.6
	Sherpa	01	2.1
	Chettri	03	6.38
Age group (years)	30–40	14	29.8
	41–50	07	14.9
	51–60	15	31.9
	61–70	08	17
	71–80	03	6.4
Education	No formal education	31	66
	Primary level	11	23.4
	Secondary level	04	8.5
	Higher education	01	2.1
Experience in herbal practice (years)	5–10	14	29.8
	11–25	14	29.8
	26–50	11	23.4
	>50	08	17
Occupation	Herbalists	11	23.5
	Farmers	13	27.6
	Daily wage workers	13	27.6
	Tea garden workers	10	21.3

**Table 2 plants-13-03505-t002:** Families, genera, and species of the recorded medicinal plants.

Plant Groups	No. of Families	No. of Genera	No. of Species
Dicotyledons	53	89	93
Monocotyledons	7	14	17
Pteridophytes	3	3	3
Gymnosperms	2	2	2
Total	65	108	115

**Table 3 plants-13-03505-t003:** List of plant species used in both monoherbal and polyherbal formulations (n = 17).

Sl. No.	Scientific Name and Voucher Number of the Ethnomedicinal Plant(s)	Family	Local Name	Part(s) Used	Disease/ Health Condition Cured	Employed in Formulation
1.	*Ageratina adenophora* (Spreng.) R.M.King and H.Rob./VBYS 15	Asteraceae	Kalijhar/Kalobanmara	Leaf	Early piles	Monoherbal
				Leaf	Cough and cold	Polyherbal
2.	*Berberis napaulensis* (DC.) Spreng./VBYS 04	Berberidaceae	Chutro	Stem	Cut and wounds	Monoherbal
				Stem	Jaundice	Polyherbal
3.	*Bergenia ciliata* (Haw.) Sternb./VBYS 05	Saxifragaceae	Pakhanbed	Stem	Toothache	Monoherbal
				Root	Bone fracture	Polyherbal
				Whole plant	Urinary tract infection (UTI)	Polyherbal
4.	*Bidens pilosa* L./VBYS 73	Asteraceae	Kuro	Leaf	Skin infection	Monoherbal
				Whole plant	Inflammation and puss formation in the skin	Polyherbal
5.	*Calotropis gigantea* (L.) W.T.Aiton /VBYS 06	Apocynaceae	Aakh	Flower	Sinusitis	Monoherbal
				Leaf	Sprain and body ache	Monoherbal
				Rhizome and leaf	Swelling	Polyherbal
6.	*Cuscuta reflexa* Roxb./VBYS 14	Convolvulaceae	Pahelolahara	Whole plant	Jaundice	Monoherbal
				Whole plant	Jaundice	Polyherbal
7.	*Elephantopus scaber* L./VBYS 25	Asteraceae	Kipujhar	Leaf	Puss formation with insect bite	Monoherbal
				Whole plant	Inflammation	Polyherbal
8.	*Gmelina arborea* Roxb. ex Sm./VBYS 150	Lamiaceae	Khamari	Bark	Food poisoning	Monoherbal
				Bark	Bone fracture	Polyherbal
9.	*Hellenia speciosa* (J.Koenig) S.R.Dutta/VBYS 210	Costaceae	Betlauri	Stem	Dysuria	Monoherbal
				Stem	Urinary tract infection (UTI)	Polyherbal
10.	*Hibiscus rosa-sinensis* L./VBYS 79	Malvaceae	Jawakusum	Flower	Kidney stone	Monoherbal
				Twig	Tearing of ligament	Polyherbal
11.	*Mallotus philippensis* (Lam.) Müll. Arg./VBYS 08	Euphorbiaceae	Sindure	Bark	Piles	Monoherbal
				Bark	Bone fracture	Polyherbal
12.	*Oroxylum indicum* (L.) Kurz/VBYS 142	Bignoniaceae	Totola	Bark	Liver problem	Monoherbal
				Seed	High blood pressure	Monoherbal
				Bark	Urinary problems	Polyherbal
13.	*Phyllanthus emblica* L./VBYS 145	Phyllanthaceae	Amala	Fruit	Mouth ulcer	Monoherbal
				Fruit	Jaundice	Polyherbal
14.	*Scoparia dulcis* L./VBYS 96	Plantaginaceae	Chinijhar	Leaf	Fever and sore throat	Monoherbal
				Leaf	Jaundice	Polyherbal
15.	*Tinospora cordifolia* (Willd.) Hook.f. and Thomson/VBYS 51	Menispermaceae	Gurjo	Leaf and stem	Diabetes and high blood pressure	Monoherbal
				Stem	Jaundice	Polyherbal
16.	*Tupistra nutans* Wall. exLindl./VBYS 65	Asparagaceae	Nakima	Flower	High blood pressure, lethargy, and diabetes	Monoherbal
				Root	Cough	Polyherbal
17.	*Pouzolzia zeylanica* (L.) Benn./VBYS 39	Urticaceae	Chipley	Leaf	Bone fracture	Monoherbal
				Stem and leaf	Bone fracture	Polyherbal

**Table 4 plants-13-03505-t004:** Plant species with multiple therapeutic uses (n = 28).

Sl. No.	Scientific Name, Family, and Voucher Number of Theethnomedicinal Plant	Local Name	Part(s) Used	Diseases/Health Conditions Cured	Employed in Formulation
1.	*Ageratina adenophora* (Spreng.) R.M.King and H.Rob. (Asteraceae)/VBYS 15	Kalijhar/Kalobanmara	Leaf	Early piles	Monoherbal.
			Leaf	Cut and wounds	Polyherbal. It is used alongside *Centella asiatica* and *Ocimum tenuiflorum.*
2.	*Berberi snapaulensis* (DC.) Spreng. (Berberidaceae)/VBYS 04	Chutro	Stem	Cut and wounds	Monoherbal.
			Stem	Jaundice	Polyherbal. It is used alongside *Cuscuta reflexa*, *Raphanus raphanistrum* subsp. *sativus*, *Centella asiatica*, *Equisetum arvense,* and *Hellenia speciosa.*
3.	*Bergenia ciliata* (Haw.) Sternb.(Saxifragraceae)/VBYS 05	Pakhanbed	Stem	Toothache	Monoherbal.
			Stem	Urinary tract infection (UTI)	Monoherbal.
			Root	Bone fracture	Polyherbal. It is applied with *Muehlenbeckia platyclada* and *Astilbe rivularis*
			Whole plant	Urinary tract infection (UTI)	Polyherbal. It is used alongside *Hellenia speciosa*
4.	*Bidens pilosa* L. (Asteraceae)/VBYS 73	Kuro	Leaf	Skin infection	Monoherbal.
			Whole plant	Puss formation in the wound	Polyherbal. It is applied with *Elephantopus scaber.*
5.	*Calotropis gigantea* (L.) W.T.Aiton (Apocynaceae)/VBYS 06	Aakh	Flower	Sinusitis	Monoherbal.
			Leaf	Sprain and body ache	Monoherbal.
			Leaf	Swelling	Polyherbal. It is applied with *Kaempferia rotunda.*
6.	*Causonis japonica* (Thunb.) Raf. (Vitaceae)/VBYS 75	Lahara	Fruit	Dry cough	Monoherbal.
			Leaf	Gastric problems	Monoherbal.
7.	*Centella asiatica* (L.) Urb.(Apiaceae)/VBYS 103	Golpatta	Leaf	Cough and cold	Polyherbal. It is used alongside *Ageratina adenophora* and *Ocimum tenuiflorum.*
			Whole plant	Jaundice	Polyherbal. It is applied with *Berberis napaulensis*, *Cuscuta reflexa*, *Raphanus raphanistrum* subsp. *sativus*, *Equisetum arvense,* and *Hellenia speciosa.*
			Leaf	Bronchitis	Polyherbal. It is used alongside *Hydrangea febrifuga* and *Ocimum tenuiflorum.*
			Leaf	Typhoid	Polyherbal. It is applied with *Drymaria cordata.*
8.	*Cuscuta reflexa* Roxb. (Convolvulaceae)/VBYS 14	Pahelolahara	Whole plant	Jaundice	Monoherbal.
			Whole plant	Jaundice	Polyherbal. It is used alongside *Oroxylum indicum*, *Terminalia chebula*, *Terminalia bellirica,* and *Phyllanthus emblica.*
			Whole plant	Jaundice	Polyherbal. It is applied with *Berberis napaulensis*, *Raphanus raphanistrum* subsp. *sativus*, *Centella asiatica*, *Equisetum arvense,* and *Hellenia speciosa.*
9.	*Cynodon dactylon* (L.) Pers.(Poaceae)/VBYS 89	Dubo	Twig	Bone fracture	Monoherbal.
			Leaf	Tonsillitis	Monoherbal.
10.	*Elephantopus scaber* L. (Asteraceae)/VBYS 25	Kipujhar	Leaf	Insect bite	Monoherbal.
			Whole plant	Puss formation in the wound	Polyherbal. It is used alongside *Bidens pilosa.*
11.	*Gmelina arborea* Roxb. ex Sm.(Lamiaceae)/VBYS 150	Khamari	Bark	Food poisoning	Monoherbal.
			Bark	Bone fracture	Polyherbal. It is applied with *Mallotus philippensis*, *Viscumarticulatum,* and *Urtica ardens.*
12.	*Hellenia speciosa* (J.Koenig) S.R.Dutta (Costaceae)/VBYS 210	Betlauri	Stem	Dysuria	Monoherbal.
			Stem	Urinary problems	Polyherbal. It is administered with *Oroxylum indicum*, *Terminalia chebula*, *Terminalia bellirica,* and *Phyllanthus emblica.*
			Stem	Urinary tract infection (UTI)	Polyherbal. It is applied with *Bergenia ciliata.*
			Stem	Urinary tract infection (UTI)	Polyherbal. It is administered with *Musa acuminata*
			Stem	Urinary tract infection (UTI)	Polyherbal. It is used alongside *Sida acuta* and *Saccharum officinarum.*
			Stem	Jaundice	Polyherbal. It is applied with *Berberis napaulensis*, *Cuscuta reflexa*, *Raphanus raphanistrum* subsp. *sativus*, *Centella asiatica,* and *Equisetum arvense.*
13.	*Hibiscus rosa-sinensis* L. (Malvaceae)/VBYS 79	Jawakusum	Flower	Kidney stone	Monoherbal.
			Twig	Tearing of ligament	Polyherbal. It is used alongside *Allium sativum* and *Curcuma longa.*
14.	*Mallotus philippensis* (Lam.) Müll. Arg.(Euphorbiaceae)/VBYS 08	Sindure	Bark	Piles	Monoherbal.
			Bark	Bone fracture	Polyherbal. It is applied with *Viscumarticulatum*, *Gmelina arborea,* and *Urtica ardens.*
15.	*Muehlenbeckia platyclade* (F.Muell.) Meisn.(Polygonaceae) /VBYS 106	Bhuiharchur	Leaf and stem	Bone fracture	Polyherbal. It is administered with *Pouzolziazeylanica*.
			Leaf and stem	Bone fracture	Polyherbal. It is used alongside *Astilbe rivularis* and *Bergenia ciliata.*
16.	*Oroxylum indicum* (L.) Kurz (Bignoniaceae)/VBYS142	Totola	Bark	Liver problems	Monoherbal.
			Seed	High blood pressure	Monoherbal.
			Fruit	Jaundice	Polyherbal. It is applied with *Cuscuta reflexa*, *Terminalia chebula*, *Terminalia bellirica,* and *Phyllanthus emblica.*
			Bark	Urinary problems	Polyherbal. It is administered with *Hellenia speciosa*, *Terminalia chebula*, *Terminaliabellirica,* and *Phyllanthus emblica.*
17.	*Phyllanthus emblica* L. (Phyllanthaceae)/VBYS 145	Amala	Fruit	Mouth ulcer	Monoherbal.
			Fruit	Jaundice	Polyherbal. It is applied with *Cuscuta reflexa*, *Oroxylum indicum*, *Terminalia chebula,* and *Terminalia bellirica.*
			Fruit	Urinary problems	Polyherbal. It is used alongside *Oroxylum indicum*, *Hellenia speciosa*, *Terminaliabellirica,* and *Terminalia chebula.*
			Fruit	Gastric problems and high blood pressure	Polyherbal. It is applied with *Terminalia bellirica* and *Terminalia chebula.*
18.	*Plantago asiatica* L. (Plantaginaceae)/VBYS 91	Camchijhar/ Bhringaraj	Root	Throat pain	Monoherbal.
			Leaf	Eye problem	Monoherbal.
19.	*Potentilla fulgens* Wall. ex Sims (Rosaceae)/VBYS 38	Banmula	Leaf	Warts	Monoherbal.
			Root	Blood dysentery and piles	Monoherbal.
20.	*Psidium guajava* L. (Myrtaceae)/VBYS 35	Ambak	Bark	Diarrhea	Monoherbal.
			Leaf	Diarrhea	Monoherbal.
21.	*Raphanus raphanistrum* subsp. sativus (Brassicaceae)/VBYS 55	Mula	Root	Jaundice	Polyherbal. It is used alongside *Cucumis sativus* and *Tinospora cordifolia.*
			Seeds	Jaundice	Polyherbal. It is applied with *Berberis napaulensis*, *Cuscuta reflexa*, *Centella asiatica,* and *Equisetum arvense.*
22.	*Rhododendron arboreum* Sm. (Ericaceae)/VBYS 521	Laliguras	Flower	Blood dysentery	Monoherbal.
			Flower	Fish bone stuck in the throat	Monoherbal.
23.	*Rumexnepalensis* Spreng. (Polygonaceae)/VBYS 40	Halhale	Root	Skin diseases and scabies	Monoherbal.
			Leaf	Piles	Monoherbal.
24.	*Scoparia dulcis* L. (Plantaginaceae)/VBYS 96	Chinijhar	Leaf	Fever and sore throat	Monoherbal.
			Leaf	Jaundice	Polyherbal. It is used alongside *Sida acuta* Burm.f. and *Elettaria cardamomum.*
25.	*Terminalia bellirica* (Gaertn.) Roxb. (Combretaceae)/VBYS 171	Barra	Fruit	Urinary problems	Polyherbal. It is administered with *Oroxylum indicum*, *Hellenia speciosa,* and *Terminalia chebula.*
			Fruit	Gastric problem and high blood pressure	Polyherbal. It is applied with *Terminalia chebula* and *Phyllanthus emblica.*
			Fruit	Jaundice	Polyherbal. It is applied with *Cuscuta reflexa*, *Oroxylum indicum*, *Terminalia chebula,* and *Phyllanthus emblica.*
26.	*Terminalia chebula* Retz. (Combretaceae)/VBYS 189	Harra	Fruit	Urinary problems	Polyherbal. It is used alongside *Oroxylum indicum*, *Hellenia speciosa*, *Terminalia bellirica,* and *Phyllanthus emblica.*
			Fruit	Gastric problems and high blood pressure	Polyherbal. It is administered with *Terminalia bellirica*, *Terminalia chebula,* and *Phyllanthus emblica.*
27.	*Tinosporacordifolia* (Willd.) Hook.f. and Thomson (Menispermaceae)/VBYS 51	Gurjo	Leaf and stem	Diabetes and high blood pressure	Monoherbal.
			Stem	Jaundice	Polyherbal. It is used alongside *Cucumis sativus* and *Raphanus raphanistrum* subsp. *sativus.*
28.	*Tupistra nutans* Wall.ex Lindl. (Asparagaceae)/VBYS 65	Nakima	Flower	High blood pressure, lethargy and diabetes	Monoherbal.
			Root	Cough	Polyherbal. It is applied with *Curcuma longa* and *Piper nigrum.*

**Table 5 plants-13-03505-t005:** List of ethnomedicinal plant species used for preparing the monoherbal formulations (n = 91).

Sl. No.	Scientific Name, Family, and Voucher Number of the Ethnomedicinal Plant (s)	Local Name	Part(s) Used	Disease/Health Condition Cured	Method of Preparation of Crude Drug and Dose Regime	Mode of Administration of the Crude Drug	UV	FL (%)
1.	*Achyranthes aspera* L. (Amaranthaceae)/VBYS 72	Akhleyjhar	Root	Labor pain	The roots are crushed, and juice is extracted. Then the juice is mixed with palm candy. A total of 1–2 teaspoons are taken after food until pain is reduced.	Oral	0.68	54.2
2.	*Acorus calamus* L. (Acoraceae) /VBYS 01	Bojo	Root	Stomach problems	The roots are cut into small pieces. One to two pieces of root are taken twice a day until the ailment is cured.	Oral	0.51	81
3.	*Aconitum heterophyllum* Wall. ex Royle (Rananculaceae)/VBYS 02	Bikuma (kalo)	Root	Food poisoning	The roots (2–3 cm) are cut into pieces. One to two pieces of roots are taken once a day for three days.	Oral	0.23	45.5
4.	*Ageratina adenophora* (Spreng.) R.M.King and H.Rob. (Asteraceae) /VBYS 15	Kalijhar/Kalobanmara	Leaf	Early piles	The juice is extracted from the leaves. One teaspoon of juice is taken twice a day after food for 1 month.	Oral	0.64	68.5
5.	*Allium rhabdotum* Stearn (Amaryllidaceae) /VBYS 03	Jimbu	Leaf	High blood pressure	The leaves are crushed with salt and made into a paste. One to two teaspoons of paste is taken after a meal for 3 days.	Oral	0.32	56.7
6.	*Berberis napaulensis* (DC.) Spreng. (Berberidaceae) /VBYS 04	Chutro	Stem	Cuts and wounds	The stem is made into a paste and applied to cuts and wounds until they are cured properly.	Topical	0.44	56.3
7.	*Bergenia ciliata* (Haw.) Sternb. (Saxifragaceae) /VBYS 05	Pakhanbed	Stem	Toothache	About 5–6 stems are taken and cut into small pieces. From that, the stem paste is prepared. The paste is applied to the teeth every day until the toothache has subsided.	Topical	0.68	65.2
			Stem	Urinary problem	The stem juice is extracted. A total of 1–2 cups of juice is taken twice a day after food for 15 days.	Oral		
8.	*Betula alnoides* Buch.-Ham. ex D.Don (Betulaceae) /VBYS 71	Saur	Bark	Snakebites cuts and wounds, inflammation	The fresh bark is ground with water and made into a paste. Then it is applied as a plaster and changed after 3 days.	Topical	0.88	97
9.	*Bidens pilosa* L. (Asteraceae) /VBYS 73	Kuro	Leaf	Skin infection	The leaves are crushed, and leaf juice is applied to the infected area for 7 days.	Topical	0.58	37
10.	*Brugmansia suaveolens* (Humb. and Bonpl. ex Willd.) Sweet (Solanaceae) /VBYS 74	Ghanti ful	Leaf	Muscular pain	Fresh leaves are heated over a low flame and placed on the affected area. This process is repeated twice a day until the ailment is cured.	Topical	0.48	51.2
11.	*Calotropis gigantea* (L.) W.T.Aiton (Apocynaceae) /VBYS 06	Aakh	Flower	Sinusitis	A handful of flowers are boiled in water. The steam is inhaled before sleeping to cure sinusitis.	Inhalation	0.71	82.3
			Leaf	Sprains, body aches	The leaves are heated over a low flame. The heated leaves are placed on the affected area and kept there for 3–5 min. This procedure is repeated 3 times.	Topical		
12.	*Carica papaya* L. (Caricaceae) /VBYS 223	Mewa	Leaf	Dengue	The leaf decoction is prepared. One cup of decoction is taken after every meal until the disease is cured.	Oral	0.35	82.3
13.	*Catharanthus roseus* (L.) G.Don (Apocynaceae) /VBYS 07	Sadabahar	Leaf and flower	Diabetes, high blood pressure	Two to three flowers or two to three leaves are taken directly every morning.	Oral	0.41	45.2
14.	*Causonis japonica* (Thunb.) Raf. (Vitaceae) /VBYS 75	Lahara	Fruit	Dry cough	The fruits (2–3) are eaten twice a day until the cough is cured.	Oral	0.79	57.2
			Leaf	Gastric problems	The leaf decoction is taken twice a day after food for 2–3 days.	Oral		
15.	*Chromolaena odorata* (L.) R.M.King and H.Rob. (Asteraceae) /VBYS 76	Seto banmara	Leaf	Cuts and wounds	Fresh leaves are crushed and made into a paste. Then the paste is applied externally on cuts and wounds until they are cured.	Topical	0.58	45.6
16.	*Cinnamomum tamala* (Buch.-Ham.) T. Nees and C.H.Eberm. (Lauraceae) /VBYS 115	Tejpatta	Bark	Diarrhea	The decoction is prepared from bark. One to two teaspoons of decoction are taken twice a day after food for 3 days.	Oral	0.23	57.4
17.	*Cinnamomum verum* J.Presl (Lauraceae) /VBYS 09	Sinkauli	Bark	Cough	The decoction is prepared from bark. Two teaspoons of decoction are taken twice a day until the ailment is cured.	Oral	0.46	56.2
18.	*Clerodendrum thomsoniae* Balf.f. (Lamiaceae) /VBYS 11	Seto ratoful	Leaf	Viral fever	The leaf juice is extracted. A total of 1–2 teaspoons of juice is taken after every meal until the disease is cured.	Oral	0.11	72.2
19.	*Curcuma caesia* Roxb. (Zingiberaceae) /VBYS 13	Kalo hardi/Haledo	Rhizome	Gastric problems	The rhizome is dried and made into powder. A total of 1–2 teaspoons of powder is taken with 1 cup of water after food for 2–3 days.	Oral	0.54	84
20.	*Cuscuta reflexa* Roxb. (Convolvulaceae) /VBYS 14	Pahelo lahara	Whole plant	Jaundice	The whole plant is crushed, and juice is extracted. One cup of juice is consumed twice a day until the illness is cured.	Oral	0.41	49.9
21.	*Cymbopogon citratus* (DC.) Stapf (Poaceae) /VBYS 16	Lemon grass	Leaf	Liver complaints	The leaf juice is extracted. One cup of leaf juice is taken on an empty stomach until the ailment is cured.	Oral	0.42	52.8
22.	*Cynodon dactylon* (L.) Pers. (Poaceae) /VBYS 89	Dubo	Twig	Bone fracture	Twigs are ground and made into a paste. For 7 days, the paste is applied as a plaster.	Topical	0.84	75.2
			Leaf	Tonsillitis	After 7 days, fresh paste is applied. The leaves are crushed, and juice is extracted. A total of 1–2 teaspoons of this juice is consumed in the morning and evening after meals until the ailment is cured.	Oral		
23.	*Cynoglossum zeylanicum* (Sw. ex Lehm.) Thunb. ex Brand (Boraginaceae) /VBYS 17	Lampatey	Root and leaf	Burns	A paste is prepared from both the roots and leaves. Every day, a fresh application of the paste is performed on the affected part.	Topical	0.46	56.8
24.	*Dactylicapnos scandens* (D.Don) Hutch.(Papaveraceae) /VBYS 20	Ashwalata	Whole plant	Heart problems	The entire plant is dried. About 250 g of the whole plant is taken, and a decoction is prepared with honey. One teaspoon is consumed in the morning and evening.	Oral	0.44	66.8
25.	*Datura stramonium* L. (Solanaceae) /VBYS 119	Dhatura	Seed	Rabies	First, the seed paste is prepared. Then, it is applied to the affected area as a plaster for 5 days. Subsequently, after the initial 5 days, the plaster is replaced.	Topical	0.67	58.4
26.	*Dracaena trifasciata* (Prain) Mabb. (Asparagaceae) /VBYS 23	Snake plant	Root	Snake bite	The root paste is prepared and applied to the affected area as a plaster for a duration of 3 days. Following this period, the plaster is replaced.	Topical	0.72	75
27.	*Elaeocarpus angustifolius* Blume (Elaeocarpaceae) /VBYS 24	Rudraksh	Bark	Diabetes	About 150 g of bark decoction is prepared in 1 liter of water. The barks are collected in the morning and kept under sunlight for few hours before preparing the decoction. A total of 1–2 teaspoons are taken in the morning on an empty stomach.	Oral	0.8	58.9
28.	*Elephantopus scaber* L. (Asteraceae) /VBYS 25	Kipu jhar	Leaf	Insect bite	About 10–15 leaves are crushed and made into a paste. The leaf paste is taken for 2–3 days.	Topical	0.51	67.2
29.	*Entada gigas* (L.) Fawc. and Rendle (Fabaceae) /VBYS 77	Pangra	Seed	Skin infection	The seeds are made into a paste and applied to the affected area as a plaster until the ailment is cured.	Topical	0.41	56.2
30.	*Euphorbia hirta* L.(Euphorbiaceae) /VBYS 26	Chini jhar/Jirejhar	Whole plant	Black urine	The whole plant is crushed, and the juice is extracted. One small cup of juice is taken in the morning and evening after food until the ailment is cured.	Oral	0.48	71.2
31.	*Ginkgo biloba* L. (Ginkgoaceae) /VBYS 78	Gingko	Leaf	Cancer	The leaves are crushed, and the juice is extracted. A total of 1–2 teaspoons of juice is taken after food twice a day.	Oral	0.42	52.3
32.	*Gmelina arborea* Roxb. ex Sm. (Lamiaceae) /VBYS 150	Khamari	Bark	Food poisoning	The bark (250 g) decoction is prepared in 500 mL of water. A total of 1–2 teaspoons of decoction is taken twice a day after meals until the illness is cured.	Oral	0.26	80.1
33.	*Hellenia speciosa* (J.Koenig) S.R.Dutta (Costaceae) /VBYS 210	Betlauri	Stem	Urinary problems	The stems are crushed, and the juice is extracted. One glass of juice is taken on an empty stomach every morning.	Oral	0.97	61.2
34.	*Hibiscus rosa-sinensis* L. (Malvaceae) /VBYS 79	Jawa kusum	Flower	Kidney stone	The flowers are made into a paste. One teaspoon is taken on an empty stomach in the morning.	Oral	0.59	55.5
35.	*Houttuynia cordata* Thunb. (Saururaceae) /VBYS 53	Fapharjhar/Gandeyjhar	Whole plant	Jaundice	The juice is extracted from the plant. One to two teaspoons of juice are taken in the morning and evening after food until the illness is cured.	Oral	0.42	75.4
36.	*Hedychium spicatum* Sm. (Zingiberaceae) /VBYS 80	Cacoush/Patnawlo	Whole plant	Tonsillitis	The whole plant is crushed, and the juice is extracted. One to two teaspoons of juice are taken twice a day until the tonsillitis is cured.	Oral	0.42	58.4
37.	*Hydrocotyle himalaica* P.K.Mukh. (Araliaceae) /VBYS 27	Golpatta	Leaf	Urinary tract infection (UTI)	About 100 g of leaves are taken and crushed. Then, the juice is extracted. The leaf juice (1 cup) is taken in the morning on an empty stomach.	Oral	0.52	67.5
38.	*Hypericum beanii* N.Robson (Hypericaceae) /VBYS 28	Runchejhar	Leaf	Urinary problem	The leaves (250 g) are crushed, and juice is extracted. One to two teaspoons of juice are taken every morning.	Oral	0.43	72.3
39.	*Kalanchoe pinnata* (Lam.) Pers. (Crassulaceae) /VBYS 30	Patharkatta/Baklopatta/Chawrase	Leaf	Urinary tract infection(UTI), kidney stone	One and a half leaves are taken on an empty stomach daily.	Oral	0.62	67.8
40.	*Kaempferia galanga* L.(Zingiberaceae) /VBYS 81	Facheng	Rhizome	Gastric problems	The rhizome is chewed during gastric problems.	Oral	0.41	56.2
41.	*Leucosceptrumcanum* Sm. (Lamiaceae) /VBYS 31	Ghurpis	Flower	Cough	One to two flowers are taken twice a day on an empty stomach for a week.	Oral	0.21	57.8
42.	*Litsea cubeba* (Lour.) Pers. (Lauraceae) /VBYS 125	Siltimbur	Leaf	Mouth soar	One to two leaves are chewed every day until the disease is cured.	Oral	0.45	65.8
43.	*Lycopodium japonicum* Thunb.(Lycopodiaceae) /VBYS 32	Nagbeli	Root	Stomach troubles	Dried roots are pounded and made into a powder. One teaspoon of root powder is taken with water in the morning on an empty stomach.	Oral	0.22	75.2
44.	*Mallotus philippensis* (Lam.) Müll. Arg. (Euphorbiaceae) /VBYS 08	Sindure	Bark	Piles	About 100 g of bark is ground and boiled with water, honey, and egg. One to two teaspoons are taken in the morning and evening.	Oral	0.45	29
45.	*Malvaviscus penduliflorus* Moc. and Sessé ex DC. (Malvaceae) /VBYS 82	Chusneful/Lurkeyful	Root	Pneumonia	The root juice is extracted. One to two teaspoons of juice are taken twice a day after food until the illness is cured.	Oral	0.38	42
46.	*Melia azedarach* L. (Meliaceae) /VBYS 83	Bagaina	Leaf	High blood pressure	One teaspoon of leaf decoction is taken once a day after a meal.	Oral	0.44	48.9
47.	*Mimosa pudica* L. (Fabaceae) /VBYS 33	Buhari jhar	Root	Toothache	The roots are dried and made into powder. A small amount of powder is taken and applied to the affected area until the pain has subsided.	Topical	0.16	56.4
48.	*Mirabilis jalapa* L. (Nyctaginaceae) /VBYS 84	Chandan ful	Flower and seed	Freckles	The fresh flowers and seeds are made into a paste and applied to the skin every night.	Topical	0.37	44.2
49.	*Moringaoleifera* Lam. (Moringaceae) /VBYS 34	Sajana	Tender leaf	High blood pressure	A handful of tender leaves are taken. A decoction is prepared. A small cup of decoction is taken once a day after food.	Oral	0.21	64.2
50.	*Mussaenda treutleri* Stapf. (Rubiaceae) /VBYS 66	Sitalu	Bark and root	Heart problems	The roots and bark are washed and made into a paste. About 50 g of the paste is taken once a day after breakfast.	Oral	0.26	56.4
51.	*Nyctanthes arbor-tristis* L. (Oleaceae) /VBYS 10	Parijat	Leaf	High blood pressure, Diabetes	One teaspoon of leaf decoction is taken once a day on an empty stomach.	Oral	0.47	61.2
52.	*Nephrolepis cordifolia* (L.) C.Presl (Polypodiaceae) /VBYS 48	Pani amala	Rhizome	Blood circulation, Kidney disease	The rhizome is washed and consumed directly once a day in the morning on an empty stomach	Oral	0.51	58.5
53.	*Ocimum basilicum* L. (Lamiaceae) /VBYS 87	Babari ful	Leaf	Skin rashes	The leaf juice and leaf paste are applied to the affected area of the skin.	Topical	0.43	52
54.	*Oroxylum indicum* (L.) Kurz (Bignoniaceae) /VBYS 142	Totola	Bark	Liver problems	In 500 mL of water, 100 g of bark is boiled and left overnight. One to two teaspoons of the decoction is taken at night after dinner daily.	Oral	0.45	68.2
			Seed	High blood pressure	The seeds are roasted and made into a powder, and one spoon is taken per day with water.	Oral		
55.	*Oxalis corniculata* L.(Oxalidaceae) /VBYS 130	Chari amilo	Leaf	Cataract and eye problem	Leaf juice is made using a handful of leaves. The leaf juice is administered to the eyes once a day at night until the ailment is healed properly.	Topical	0.12	52.1
56.	*Passiflora edulis* Sims (Passifloraceae) /VBYS 88	Garandal	Seed	Stomach problems	The seeds are taken directly.	Oral	0.35	54.1
57.	*Phytolacca acinosa* Roxb. (Phytolaccaceae) /VBYS 18	Jaringo	Root	Skin psoriasis	The roots are made into a paste and applied to the affected area as a plaster.	Topical	0.32	54.2
58.	*Phyllanthus emblica* L. (Phyllanthaceae) /VBYS 145	Amala	Fruit	Mouth ulcer	One to two fruits are taken once a day.	Oral	0.71	68.2
59.	*Phyllanthus urinaria* L. (Phyllanthaceae) /VBYS 90	Bhuiamala	Fruit	Diabetes	The fruits are directly taken once a day.	Oral	0.52	55.6
60.	*Plantago asiatica* L. (Plantaginaceae) /VBYS 91	Camchijhar/ Bhringaraj	Root	Throat pain	The roots are washed, crushed, and formed into a paste. One teaspoon of this paste is consumed twice daily.	Oral	0.78	87
			Leaf	Eye problems	The leaves are crushed, and juice is extracted and applied to the affected area.	Topical		
61.	*Psidium guajava* L. (Myrtaceae) /VBYS 35	Ambak	Bark	Diarrhea	The bark (50 g) is made into a paste. The bark paste is taken directly before food.	Oral	0.56	49.6
			Leaf	Diarrhea	About 7–9 tender leaves are chewed until the disease is cured.	Oral		
62.	*Piper retrofractum* Vahl (Piperaceae) /VBYS 36	Chabo/Chaba	Root	Cold and cough	The roots are dried and cut into small pieces. One to two pieces are consumed until the cough is cured.	Oral	0.21	50
63.	*Piper longum* L. (Piperaceae) /VBYS 37	Pipla	Fruit	Cough and kidney problems	One to two fruits are taken in the morning and evening after meals.	Oral	0.45	54.1
64.	*Potentilla fulgens* Wall. ex Sims (Rosaceae) /VBYS 38	Banmula	Leaf	Warts	Juice is prepared from the leaves. The warts are washed with potassium, and leaf juice is applied until they are cured.	Topical	0.84	63.5
			Root	Dysentery and piles	The root powder (250 g) and (100 g) jaggery is boiled and covered with cheesecloth. One to two teaspoons of the decoction is taken twice a day after food until the disease is cured.	Oral		
65.	*Pouzolzia zeylanica* (L.) Benn. (Urticaceae) /VBYS 39	Chipley	Leaf	Bone fracture	The leaves are made into a paste. The leaf paste is applied as a plaster for 1 week.	Topical	0.22	81.2
66.	*Pseudognaphalium affine* (D.Don) Anderb. (Asteraceae) /VBYS 92	Galeneyjhar/Boquetful	Leaf and root	Diabetes	The leaves and roots are made into a paste. One to two teaspoons of the paste are taken once a day.	Oral	0.44	35.2
67.	*Rhododendron arboreum* Sm. (Ericaceae) /VBYS 521	Laliguras	Flower	Dysentery	Two to three flowers are taken, and a decoction is prepared. The decoction is taken twice a day after food.	Oral	0.75	52.1
			Flower	Fish bone stuck in the throat.	Dried flowers (1–2) are taken immediately.	Oral		
68.	*Rhus chinensis* Mill. (Anarcardiaceae) /VBYS 240	Bhakmilo	Fruit	Dysentery	The fruits are boiled in water until a thick decoction called “chuk” is obtained. One teaspoon of this decoction is taken twice a day after meals until the illness is cured.	Oral	0.54	63
69.	*Rauvolfia serpentina* (L.) Benth. ex Kurz (Apocynaceae) /VBYS 93	Sarpagandha	Root	Snakebite	The roots are made into a paste and applied to the affected area as a plaster for 3 days. After 3 days, a fresh paste is applied again and left for the next 3 days.	Topical	0.76	69.7
70.	*Rubus ellipticus* Sm. (Rosaceae) /VBYS 94	Aiselu	Twig	Stomach problems	The handful of twigs is consumed directly twice a day until the illness is cured.	Oral	0.47	55.1
71.	*Rumex nepalensis* Spreng. (Polygonaceae) /VBYS 40	Halhale	Root	Skin diseases and scabies	The root paste is applied to the affected part as a plaster and left for 5 days. If the ailment is not cured within 5 days, a new paste may be used.	Topical	0.74	52.1
			Leaf	Piles	The leaf juice is extracted. One teaspoon of juice is taken once a day after food.	Oral		
72.	*Salvia splendens* Sellow ex Nees (Lamiaceae) /VBYS 95	Ratoful	Leaf	Mouth ulcer	The leaves are taken directly twice a day until the mouth ulcer is cured.	Oral	0.37	44.2
73.	*Scadoxus multiflorus* (Martyn) Raf. (Amaryllidaceae) /VBYS 41	Fudbali	Bulb	Mumps	The bulbs are made into a paste and applied to the affected area as a plaster for 7 days.	Topical	0.02	49.2
74.	*Schima wallichii* (DC.) Korth. (Theaceae) /VBYS 86	Chilawne	Bark	For cracked skin and feet	The bark is made into a paste. The paste is applied every night.	Topical	0.45	77.4
75.	*Scoparia dulcis* L. (Plantaginaceae) /VBYS 96	Chini jhar	Leaf	Fever, sore throat	One teaspoon of the leaf juice is taken twice a day after meals until the disease is cured.	Oral	0.38	34.2
76.	*Senna alata* (L.) Roxb. (Fabaceae) /VBYS 97	Paheloful	Leaf	Ringworm	The leaf juice is applied to the affected area twice a day.	Topical	0.38	49.8
77.	*Shorea robusta* C.F.Gaertn. (Dipterocarpaceae) /VBYS 42	Sakhuwa	Bark	Blood dysentery	The bark (100 g) and two teaspoons of honey are boiled with water. One to two teaspoons of decoction are taken in the morning and evening till the ailment is cured.	Oral	0.49	56.7
78.	*Smallanthus sonchifolius* (Poepp.) H.Rob. (Asteraceae) /VBYS 98	Bhui sew	Leaf	Heart problems	The leaves are dried and made into a powder. One teaspoon of powder is taken with hot/cold water once a day after food.	Oral	0.51	52.3
79.	*Stephania rotunda* Lour. (Menispermaceae) /VBYS 43	Ghantetamarke	Root	Gastric problems	The roots are dried and ground and made into a powder. One teaspoon of powder is taken with water in the morning and at night for 3 days.	Oral	0.58	64.2
80.	*Syzygium aromaticum* (L.) Merr. and L.M.Perry (Myrtaceae) /VBYS 44	Lwang	Flower bud	Toothache	One to two dried flower buds are taken and placed in the teeth and kept for 20–30 min until the pain has subsided.	Topical	0.45	61.9
81.	*Syzygium cumini* (L.) Skeels (Myrtaceae) /VBYS45	Jamuna	Fruit	Diabetes	Two to three fruits are taken in raw form every day.	Oral	0.47	74.1
82.	*Tagetes erecta* L. (Asteraceae) /VBYS 109	Sayapatri	Flower	Pneumonia	One to two teaspoons of flower juice are taken twice a day until the disease is cured.	Oral	0.65	69.2
83.	*Taxus baccata* L. (Taxaceae) /VBYS 100	Dhangreysalla	Bark	Cancer	One to two teaspoons of bark decoction is taken twice a day after meals.	Oral	0.61	69.8
84.	*Tectaria coadunata* (J.Sm.) C.Chr. (Polypodiaceae) /VBYS 101	Masi nigure	Root	Dysentery	The roots are crushed, and juice is extracted. A total of 1–2 teaspoons of juice is taken on an empty stomach for 3–4 days.	Oral	0.58	59.6
85.	*Terminalia arjuna* (Roxb. ex DC.) Wight and Arn. (Combretaceae) /VBYS 47	Arjun	Bark	Heart problems	The bark is ground, and a decoction is prepared. One cup of decoction is taken 2–3 times a day after food until the disease is cured.	Oral	0.62	67.2
86.	*Tinospora cordifolia* (Willd.) Hook.f. and Thomson (Menispermaceae) /VBYS 51	Gurjo	Leaf and stem	Diabetes, high blood pressure	The leaves and stems are cut into small pieces. Then, a decoction is prepared. One cup of decoction is taken in the morning on an empty stomach every day.	Oral	0.74	59.9
87.	*Tridax procumbens* L. (Asteraceae) /VBYS 102	Thareyjhar	Whole plant	Diarrhea and dysentery	The whole plant is crushed, and juice is extracted. One to two teaspoons of juice are taken twice a day until the ailments are cured.	Oral	0.46	47
88.	*Tropaeolum majus* L. (Tropaeolaceae) /VBYS 120	Paheloful lahara	Twig	Stomach problems	The twigs are crushed, and juice is extracted. One to two teaspoons of juice are taken three times a day until the disease is cured.	Oral	0.39	48.6
89.	*Tupistra nutans* Wall.ex Lindl. (Asparagaceae) /VBYS 65	Nakima	Flower	High blood pressure, lethargy, diabetes	The flower is boiled and made into a paste. One teaspoon of the paste is taken once a day every morning.	Oral	0.53	56.
90.	*Youngia japonica* (L.) DC. (Asteraceae) /VBYS 104	Mula pate	Leaf	Burns	The leaf paste is applied to the affected area as plaster and changed every day.	Topical	0.56	66.2
91.	*Zanthoxylum acanthopodium* DC. (Rutaceae) /VBYS 105	Boke timbur	Fruit	Toothache	The fruit is made into a paste and applied to the teeth until the pain has subsided.	Topical	0.67	62.5

**Table 6 plants-13-03505-t006:** List of polyherbal formulations and ethnomedicinal plant species used (n = 21).

Sl. No.	Scientific Name, Family, and Voucher Number of the Ethnomedicinal Plant (s)	Local Name	Part(s) Used	Disease /Health Condition Cured	Mode of Preparation of Crude Drug and Dose Regime	Modeof Administration of Crude Drug
1.	*Ageratina adenophora* (Spreng.) R.M.King and H.Rob. (Asteraceae) /VBYS 15	Kali jhar	Leaf	Cough and cold	The leaves of *A*. *adenophora*, *C. asiatica,* and *O. tenuiflorum* are taken in equal amounts (1:1:1), and juice is extracted. One to two teaspoons are taken every day in the morning and evening until the illness is cured.	Oral
	*Centella asiatica* (L.) Urb. (Apiaceae) /VBYS 103	Golpatta	Leaf			
	*Ocimum tenuiflorum* L. (Lamiaceae) /VBYS 114	Tulsi	Leaf			
2.	*Elephantopus scaber* L. (Asteraceae) /VBYS25	Kipu jhar	Whole plant	Puss formation in the wound.	The whole plants of *E. scaber* and *B*. *pilosa* are taken in equal amounts (1:1). Then, they are crushed, and juice is extracted. The juice is applied for 3–4 days.	Topical
	*Bidens pilosa* L. (Asteraceae) /VBYS 73	Kuro jhar	Whole plant			
3.	*Oroxylum indicum* (L.) Kurz (Bignoniaceae) /VBYS142	Totola	Fruit	Jaundice	The whole plant of *C. reflexa* and fruits of *O*. *indicum*, *T. chebula*, *P. emblica,* and *T*. *bellirica* (1:1:1:1:1) are crushed, and the juice is squeezed. One teaspoon of juice is taken 3 times a day after food till the disease is cured.	Oral
	*Terminalia chebula* Retz. (Combretaceae) /VBYS 189	Harra	Fruit			
	*Terminalia bellirica* (Gaertn.) Roxb. (Combretaceae) /VBYS 171	Barra	Fruit			
	*Phyllanthus emblica* L. (Phyllanthaceae) /VBYS 145	Amala	Fruit			
	*Cuscuta reflexa* Roxb. (Convolvulaceae) /VBYS 14	Pahelo lahara	Whole plant			
4.	*Oroxylum indicum* (L.) Kurz (Bignoniaceae) /VBYS 142	Totola	Bark	Urinary problems	The barks of *O*. *indicum*, stems of *H. speciosa,* and fruits of *T.bellirica*, *T*. *chebula,* and *P.emblica* are taken in equal amounts (1:1:1:1:1). The juice is extracted and mixed with palm candy. One to two teaspoons of juice are taken only at night after food for 10–12 days.	Oral
	*Hellenia speciosa* (J.Koenig) S.R.Dutta (Costaceae) /VBYS 210	Betlauri	Stem			
	*Terminalia bellirica* (Gaertn.) Roxb. (Combretaceae) /VBYS 171	Barra	Fruit			
	*Terminalia chebula* Retz. (Combretaceae) /VBYS 189	Harra	Fruit			
	*Phyllanthus emblica* L. (Phyllanthaceae) /VBYS 145	Amala	Fruit			
5.	*Terminalia bellirica* (Gaertn.) Roxb. (Combretaceae) /VBYS 171	Barra	Fruit	Gastric problems, High blood pressure	The dried fruits of *T*. *bellirica*, *T*. *chebula,* and *P*. *emblica* are taken in equal amounts (1:1:1) and made into powder. One teaspoon of powder is taken in the morning with lukewarm water on an empty stomach daily.	Oral
	*Terminalia chebula* Retz. (Combretaceae) /VBYS 189	Harra	Fruit			
	*Phyllanthus emblica* L. (Phyllanthaceae) /VBYS 145	Amala	Fruit			
6.	*Bergenia ciliata* (Haw.) Sternb. (Saxifragraceae) /VBYS 05	Pakhanbed	Whole plant	Urinary tract infection (UTI)	The whole plant of *B*. *ciliata* and stems of *H. speciosa* are taken in equal amounts (1:1). Juice is extracted and mixed with palm candy. One cup of juice is taken in the morning and at night until the ailment is cured.	Oral
	*Hellenia speciosa* (J.Koenig) S.R.Dutta (Costaceae) /VBYS 210	Betlauri	Stem			
7.	*Hellenia speciosa* (J.Koenig) S.R.Dutta(Costaceae) /VBYS 210	Betlauri	Stem	Urinary tract infection (UTI)	The stems of *H. speciosa* and *M. acuminata* are taken (1:1), and juice is extracted. One glass of juice is taken at night daily for 15 days.	Oral
	*Musa acuminata* Colla (Musaceae) /VBYS 50	Kera	Stem			
8.	*Cucumis sativus* L. (Cucurbitaceae) /VBYS54	Kakra	Fruit	Jaundice	The fruits of *C. sativus*, roots of *R. sativus,* and stems of *T. cordifolia* (1:3:1) are cut into pieces and mixed with the curd, and a paste is prepared. Two to three teaspoons of the paste are taken after food twice a day until the disease is cured.	Oral
	*Raphanusraphanistrum* subsp. *sativus*(Brassicaceae) /VBYS 55	Mula	Root			
	*Tinospora cordifolia* (Willd.) Hook.f. and Thomson. (Menispermaceae) /VBYS 51	Gurjo	Stem			
9.	*Hellenia speciosa* (J.Koenig) S.R.Dutta(Costaceae) /VBYS 210	Betlauri	Stem	Urinary problems	The whole plant of *S*. *acuta*, stems of *H. speciosa,* and *S. officinarum* are crushed, and juice is extracted and mixed with alum. One cup of juice is taken once a day on an empty stomach until the ailment is cured.	Oral
	*Sida acuta* Burm.f. (Malvaceae)/VBYS 99.	Kharetojhar	Whole plant			
	*Saccharum officinarum* L. (Poaceae)/VBYS 57	Kalo ukhu	Stem			
10.	*Mallotus philippensis* (Lam.) Müll. Arg. (Euphorbiaceae) /VBYS 08	Sindure	Bark	Bone fracture	The bark of *M*. *philippensis* and *G*. *arborea*, whole plant of *V. articulatum* and, and roots of *U. ardens* are taken in equal amounts (1:1:1:1) and made into a paste. The paste is applied to the affected area as plaster for 7 days. Then, it is replaced after 7 days.	Topical
	*Viscumarticulatum* Burm.f. (Santalaceae) /VBYS 58	Harchur	Whole plant			
	*Gmelina arborea* Roxb. ex Sm. (Lamiaceae) /VBYS 150	Khamari	Bark			
	*Urtica ardens* Link (Urticaceae) /VBYS 197	Sisnu	Root			
11.	*Kaempferia rotunda* L. (Zingiberaceae) /VBYS 121	Bhuichampa	Rhizome	Bone fracture	The rhizomes of *K. rotunda* and C. *longa* are taken in a 1:1 ratio and made into a paste with limestone. The paste is applied as plaster and changed after 4–7 days.	Topical
	*Curcuma longa* L. (Zingiberaceae) /VBYS 85	Besar	Rhizome			
12.	*Scoparia dulcis* L. (Plantaginaceae) /VBYS 59	Setoful	Leaf	Jaundice	Juice is prepared from the leaves of *S. dulcis* and *S. acuta,* along with the fruits of *E. cardamomum* (1:1:2), and mixed with palm candy. A total of 1–2 teaspoons of juice is taken in the morning and evening until the disease is cured.	Oral
	*Sida acuta* Burm.f. (Malvaceae) /VBYS 99	Kharetojhar	Leaf			
	*Elettaria cardamomum* (L.) Maton (Zingiberaceae) /VBYS 60	Elaichi	Fruit			
13.	*Berberis napaulensis* (DC.) Spreng. (Berberidaceae) /VBYS 04	Chutro	Stem	Jaundice	The dried stems of *B. napaulensis* (100 g), dried whole plant of *C. reflexa* (100 g), dried seeds of *R. sativus* (200 g), C. *asiatica* (100 g) whole plant, dried whole plant of *E. arvense* (100 g), and dried stem of *H. speciosa* (200 g) are taken in 3 liters of water and boiled. Then, the mixture is covered with cotton or cheesecloth and sieved after 12 h and mixed with 2 teaspoons of isabgol. One cup is taken in the morning and the evening until the ailment is cured.	Oral
	*Cuscuta reflexa* Roxb. (Convolvulaceae) /VBYS 62	Pahelo lahara	Whole plant			
	*Raphanusraphanistrum* subsp. *sativus* (Brassicaceae) /VBYS 63	Mula	Seed			
	*Centella asiatica* (L.) Urb. (Apiaceae) /VBYS 103	Golpatta	Whole plant			
	*Equisetum arvense* L. (Equisetaceae) /VBYS 64	Kurkurejhar	Whole plant			
	*Hellenia speciosa* (J.Koenig) S.R.Dutta (Costaceae) /VBYS 210	Betlauri	Stem			
14.	*Tupistra nutans* Wall. ex Lindl. (Asparagaceae) /VBYS 65	Nakima	Root	Cough	*T. nutans* (200 g) root powder, 1 teaspoon of C. *longa* rhizome powder, and 2 teaspoons of *P. nigrum* powder are taken and boiled in 1 liter of water, and 50 g palm candy is added. Then, the mixture is covered with cheesecloth and later sieved. A total of 2–3 teaspoons of decoction for adults and 1 teaspoon of decoction for children is taken twice a day.	Oral
	*Curcuma longa* L. (Zingiberaceae) /VBYS 85	Besar	Rhizome			
	*Piper nigrum* L. (Piperaceae) /VBYS 61	Marich	Seed			
15.	*Cannabis sativa* L. (Cannabaceae) /VBYS 67	Ganja	Leaf	Stomach troubles	The dried seeds of *T. foenum-graecum* and *T. ammi* and the dried leaves of *C*. *sativa* are taken in equal amounts (1:1:1) and made into a powder. Then, the powder is mixed with black salt. A total of 1–2 teaspoons of powder is taken with lukewarm water twice a day after food.	Oral
	*Trigonella foenum-graecum* L.(Fabaceae) /VBYS 68	Methi	Seed			
	*Trachyspermum ammi* (L.) Sprague (Apiaceae) /VBYS 69	Jwano	Fruit			
16.	*Allium sativum* L. (Amaryllidaceae) /VBYS 70	Lasun	Bulb	Tearing of ligament	The bulb of *A. sativum*, twigs of *H. rosa-sinensis,* and rhizome of *C. longa* are taken in equal amounts (1:1:1) and made into a paste. The paste is applied and bandaged on the affected area as a plaster for 7 days.	Topical
	*Hibiscus rosa-sinensis* L. (Malvaceae) /VBYS 79	Jawa kusum	Twig			
	*Curcuma longa* L. (Zingiberaceae) /VBYS 85	Besar	Rhizome			
17.	*Muehlenbeckia platyclada* (F.Muell.) Meisn. (Polygonaceae) /VBYS 106	Bhuiharchur	Leaf and stem	Bone fracture	The leaves and stems of *M. platyclada* and *P. zeylanica* are taken in equal amounts (1:1) and made into a paste. The paste is applied to the affected area as plaster. After three days, the plaster is changed.	Topical
	*Pouzolzia zeylanica* (L.) Benn. (Urticaceae) /VBYS 39	Chipley	Stem and leaf			
18.	*Muehlenbeckia platyclada* (F.Muell.)Meisn. (Polygonaceae) /VBYS 106	Bhuiharchur	Leaf and stem	Bone fracture	The leaves and stem of *M. platyclada* and the roots of *A. rivularis* and *B. ciliata* are made into paste. The paste is applied to the fractured area as a plaster for 5 days. The bandage is changed after 5 days. The paste is also taken orally with water for 5 days.	Oral and topical
	*Astilbe rivularis* Buch.-Ham. ex D.Don(Saxifragaceae) /VBYS 46	Bhurokhati	Root			
	*Bergenia ciliata* (Haw.) Sternb. (Saxifragaceae) /VBYS 05	Pakhanbed	Root			
19.	*Hydrangea febrifuga* (Lour.) Y.De Smet and Granados (Hydrangeaceae) /VBYS 107	Pahari basak	Leaf	Bronchitis	The leaves of *H. febrifuga*, *O. tenuiflorum,* and *C. asiatica* (1:1:2) are crushed, and juice is extracted. One to two teaspoons of the juice are taken twice a day until the disease is cured.	Oral
	*Ocimum tenuiflorum* L. (Lamiaceae) /VBYS 114	Tulsi	Leaf			
	*Centella asiatica* (L.) Urb. (Apiaceae) /VBYS 103	Golpatta	Leaf			
20.	*Kaempferia rotunda* L. (Zingiberaceae) /VBYS 121	Bhuichampa	Rhizome	Swelling	The rhizome of *K. rotunda* is made into a paste and applied to the swelled area. Then, the leaf of *C. gigantia* is heated and placed on top of the applied paste. Then, plaster is put on it and left for 3 days.	Topical
	*Calotropis gigantea* (L.) W.T.Aiton (Apocynaceae) /VBYS 06	Aakh	Leaf			
21.	*Drymaria cordata* (L.)Willd. ex Schult.(Caryophyllaceae) /VBYS 108	Abijalo	Leaf	Typhoid	The leaf juice is extracted from *D. cordata* and *C. asiatica* (1:1). A total of 1–2 teaspoons of juice are taken twice a day after food until the ailment is cured.	Oral
	*Centella asiatica* (L.) Urb. (Apiaceae) /VBYS 103	Golpatta	Leaf			

**Table 7 plants-13-03505-t007:** Informant consensus factor (F_ic_) for each disease category.

Sl. No.	Ailments Category of ICPC-2	Number of Taxa (N_t_)	Number of Use Reports (N_ur_)	F_ic_ Value
1.	**Pregnancy, Childbearing, Family Planning:** Pregnancy symptoms/complaints other (W29)	2	7	0.83
2.	**Digestive:** Abdominal pain/cramps general (D01), Gastrointestinal infection (D17), Rectal bleeding (D16), Mouth/tongue/lip symptoms/complt. (D20), Teeth/gum disease (D82), Flatulence/gas/belching (D08), Diarrhea (D11), Jaundice (D13), Liver disease NOS (D97), Mumps (D71)	23	154	0.87
3.	**Cardiovascular:** Heart disease other (K84), Elevated blood pressure (K85)	18	50	0.65
4.	**Skin:** Rash generalized (S07), Laceration/cut (S18), Insect bite/sting (S12), Animal/human bite (S13), Skin infection other (S76), Burns/scald (S14), Skin color change (S08), Psoriasis (S29), Warts (S03), Scabies/other acariasis (S72), Skin texture symptom/complaint (S21)	14	46	0.71
5.	**Urological:** Urination problems Other (U05); Urinary symptom/complaint other (U29), Cystitis/Urinary infection other (U71), Urinary calculus (U95), Kidney symptoms/complaint (U14)	10	51	0.82
6.	**General and unspecified:** Fever (A03), Weakness/tiredness general (A04), Swelling (A08), Viral disease other/NOS (A77), Malignancy NOS (A79); No disease (A97), General disease NOS(A99)	12	40	0.72
7.	**Musculoskeletal:** Muscle pain (L18), Fracture: other (L76); Sprain/strain of joint NOS (L79)	8	25	0.71
8.	**Respiratory:** Cough (R05), Sinus symptoms/complains (R09), Tonsillitis acute (R76), Pneumonia (R81), Throat symptom/complaint (R21), Sneezing/nasal congestion (R07), Strep throat (R72), Acute bronchitis/bronchiolitis (R78)	12	50	0.78
9.	**Endocrine/Metabolic and Nutritional:** Diabetes insulin-dependent (T89)	8	17	0.56
10.	**Blood, Blood-Forming Organs, and Immune Mechanism:** Blood symptom/complaint (B04)	3	7	0.67
11.	**Eye:** Eye symptom/complaint other (F29), Cataract (F92)	4	15	0.78

**Table 8 plants-13-03505-t008:** Contents of phenolic compounds identified in methanolic bark extract of *B. alnoides* and their reported biological activities.

Name of the Compounds	Amount (mg/g Dry Tissue)	Reported Biological Activities
Chlorogenic acid	117.5	Antioxidant, antimicrobial, antitumor, hyperglycemia, neuroprotective, anti-inflammatory [18], antivenom [19], wound healing [20]
Sinapic acid	64.7	Antioxidant, antibacterial, anticancer, antianxiety [21], wound healing [22]
Caffeic acid	5.8	Antioxidant, anti-inflammatory, and anticancer activity [23], wound healing [24]
Coumarin	1.5	Antioxidant, anti-inflammation, antiviral, antitumor, antibacterial [25], antivenom [26], wound healing [27].
p-Coumaric acid	0.4	Antioxidant, anti-inflammatory, analgesic, antimicrobial [28], wound healing [29]
Gallic acid	0.1	Antioxidant, anti-inflammatory [30], antivenom [26], wound healing [31]

## Data Availability

All data are contained within the article.

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
