# Peer review of "Quantitative Ethnobotany of Medicinal Plants from Darjeeling District of West Bengal, India, along with Phytochemistry and Toxicity Study of *Betula alnoides* Buch.-Ham. ex D.Don bark"

_plants, 2024, doi:10.3390/plants13243505_

Round 1

Reviewer 1 Report

Comments and Suggestions for Authors

Dear sir,

the article 'Quantitative ethnobotany of medicinal plants from Darjeeling District of West Bengal, India along with phytochemistry and toxicity study of Betula alnoides Buch.-Ham. ex D.Don bark' is not easy two evaluate. In fact, it is composed of two parts: one is describing the main plant species used in Darjeeling natural medicine, and another it is the description of the phytochemical properties of one particular species, the bark of Betula alnoides. The authors join somehow the two parts, which it is not easy. I attach an annotated file with several minor corrections (spacing, etc.). 

Best

Comments on the Quality of English Language

Dear sir,

the English is ok. In the annotated file you can find my corrections.

Best

Author Response

From,                                                                                                                          

Prof. Chowdhury Habibur Rahaman

Department of Botany

Visva-Bharati University

Santiniketan,West Bengal, India.

e-mail: habibur_cr@visva-bharati.ac.in

To

The Reviewer,

Plants, MDPI.

Dear sir/madam,

I would like to thank you for this opportunity to resubmit a revised copy of our manuscriptentitled “Quantitative ethnobotany of medicinal plants from Darjeeling District of West Bengal, India along with phytochemistry and toxicity study of Betula alnoidesBuch.-Ham. ex D.Don bark” (Manuscript ID- 3142538). We, all the authors, would like to take this opportunity to express our gratitude for the positive feedback and insightful comments, which have greatly helped us to improve the quality of our manuscript.

In accordance to the comments, all changeswere madein the text and were marked up using the “Track Changes” function in the revised manuscript. Enclosed please find our revised manuscript and list of changes made point-by-point.

Hope the revised form of our manuscript will satisfy you in all respect.

Regards,

Yours sincerely,

Prof. Chowdhury Habibur Rahaman

Reviewer 1

Specific comment

  1. Line 8: “Our paper” should be changed to “This article”.

(Revised manuscript (RM) Line 8)- We have changed the word “Our paper”to “This article”.

  1. Line 11: There should be a space between two words “47informants”.

(Revised manuscript (RM) Line 11)- Space has added between the words.

  1. Line 11: There should be a space betweentwo words “11persons”.

(Revised manuscript (RM) Line 11)- Space has givenbetween the words.

  1. Line 22: The word “the” should be inserted in front of the word “present”.

(Revised manuscript (RM) Line 22):The word “the” is inserted in front of the word “present” and written as “the present”.

  1. Line 25 and line 28: There should be only one number after the decimal point.

(Revised manuscript (RM) Line 25 and line 29): We have ensured that the corrected manuscript contains only one digit after the decimal point, except for the values of ethnobotanical indices like Fic and UV, which require two digits after the decimal for representationof their exact value.

6.Line 33: In the keywords, the words should not be repeated as it is in the title.

(Revised manuscript (RM) Line34 )- The keywords of the revised manuscript are changed and written as follows: “Ethnomedicine; Darjeeling-Himalaya; Ethnobotanical indices; bark drug; HPLC; MTT assay”

  1. Line 41: This line has to be deleted.

(Revised manuscript (RM) line 41: The line has been deleted.

  1. Line 113: There should be only 1 space between two lines “…….day life. Among……”

(Revised manuscript (RM) Line 113: One space has been maintained between the two lines.

  1. Line 113-115: The line had to be rewritten “Among ,…..(6.38%).”

(Revised manuscript (RM) Line 114-117: The line has been corrected and rewritten “Among…. (6.4%).”

  1. Table 1: There should be only one number after the decimal point.

(Revised manuscript (RM) Table 1: There is only one number after the decimal point in Table 1.

  1. Line 131: The word “species” should be deleted.

(Revised manuscript (RM) Line 131):The word “species” is deleted from the text.

  1. Table 2: Change of capital letters to small letters in the Table 2 captions.

(Revised manuscript (RM) Table 2): The capital letters in the Table 2 captions have been changed to lowercase.

  1. Figure 1: Figure 1 was not visible to the reviewer so it had to be changed.

(Revised manuscript (RM) Figure 1): As per the instruction Figure 1 has been replaced with the corrected one.

  1. 14. Table 3: Table 3 caption has to be placed just above the table. The vertical lines in the table had to be removed.

(Revised manuscript (RM) Table 3: The caption has been placed just above the table. The lines of the vertical columnhave been removed from the table.

  1. Line 366: The word “Current study” had to be changed to “The present study”

(Revised manuscript (RM) Line 433: The word “Current study” has been changed to “The present study”.

16.. Line 421: The word “this” has to be replaced with “the”.

(Revised manuscript (RM) Line 488: The word “this” has been changed to “the”.

  1. Line 447: The word “was” has to been deleted.

(Revised manuscript (RM) Line 542: The word “was” has been deleted.

  1. Line 449: The word “it’s” has to be replaced with “its”.

(Revised manuscript (RM)Line 544: The word“it’s” has been replaced with “its”.

  1. Line 502: There should be only one space between two words“ the cell….”.

(Revised manuscript (RM) Line637: Only one space has been ensured between two words “ the cell..”.

  1. Line 522: The unit in capital letter “Km” should be “km”.

(Revised manuscript (RM) Line658: The unit in capital letter “Km” is changed to “km”.

  1. Line 532 and line 533: It is the first time you mention the word questionnaire. It should be mentioned before. Interviewees is the same as informants?

Answer 1- As per the journal's instructions, the ‘methodology’ section has been placed after the ‘discussion’ and before the ‘conclusion’, that is why the word 'questionnaire' appears towards the end of the manuscriptin methodology part. However, following your suggestion, we have now mentioned the word 'questionnaire' in the ‘introduction’as well as ‘results’section of our revised manuscript.

(Revised manuscript (RM) Line 76-80):  “Therefore,…………….questionnaire.”

(Revised manuscript (RM) Line 103-105):  “Information…….standard questionnaire.”

Answer 2:Yes, the terms 'interviewees' and 'informants' are used interchangeably in the context of this researchwork.

  1. Line 622: There should be space between “B.alnoides”.

(Revised manuscript (RM)Line 757): The space has been added.

  1. Line 624: There should be maintenance of space between “The stock…”.

(Revised manuscript (RM) Line 759-765): We ensure consistent spacing between two words.

  1. Line 642: The spelling of “standrads” is incorrect.

(Revised manuscript (RM) Line 777): The spelling has been corrected.

  1. Line 649: The sentence “104 cells ….” has to be rewritten.

(Revised manuscript (RM) Line 784-785): The line has been rewritten

26.. Line 748: The space should be maintainedbetween the two lines in reference number 5.

Revised manuscript (RM)Reference number 5: The spacehas been  maintainedbetween the two lines.

  1. Line 752: One space has to be maintained between two words “… Himalaya.Nat….”in reference number 7.

Revised manuscript (RM) Reference number 7: Space has been added between the words 'Himalaya.' and 'Nat.' in reference number 7.

  1. Line 754: One space has to be maintained between two words “9.Yonzone.” in reference number 9.

Revised manuscript (RM)Reference number 9: A space has been maintained between the words '9.' and 'Yonzone…’ in reference number 9.

  1. Line 755: One space has to be maintained between two words “10.Subba…”in reference number 10.

Revised manuscript (RM)Reference number 10.:One space gap has been maintained between two words “10.” and “Subba”.

  1. Line 765: There should be space between “Ethnopharmacol.2004…” in reference number 14.

Revised manuscript (RM)Reference number 14: Space has been added.

  1. Line 774: The should be space between “19.Toyama…”in reference number 19.

Revised manuscript (RM)Reference number 19.:Space has been added.

  1. Line 823: The word “Zingiber” should be in italicsin reference number 39.

Revised manuscript (RM)Reference number 39: The word “Zingiber” is italicised.

Reviewer 2 Report

Comments and Suggestions for Authors

In the manuscript “Quantitative ethnobotany of medicinal plants from Darjeeling District of West Bengal, India along with phytochemistry and toxicity study of Betula alnoides Buch.-Ham. ex D.Don bark”, Subba et al try to document ethnomedicinal information of 11 villages under three subdivisions of Kurseong, Darjeeling Sadar and Mirik in Darjeeling district. The justification of the study relies on a rich repository of plant species including medicinal plants in the district and the lack of etnopharmacological analysis using quantitative tools and stats. The authors documented appropriately phytotherapeutics used by the folk communities in this district. Then, Subba et al focused on Betula alnoides Buch-Ham.ex D Don, carrying out TPC, flavonoids and HPLC analysis. They chose this plant given that it was found as the most significant specie used for ethnobotanical purposes. This is the weakest section of the manuscript. The authors claim that Betula alnoides Buch-Ham.ex D Don is an “underexplored plant as a potential source of therapeutically important chemical compounds”. However, there is already published literature of this plants (see doi: 10.1080/13880209.2023.2292261; DOI: 10.3109/08923973.2012.661739, among other papers).     Major   1.   Table 8 is not fully readeable. 2.         The authors showed HPLC chromatogram obtained from methanolic bark extract of B. alnoides. Does the bark paste of this plant (topically applied for treating snakebite, inflammation, cut and wounds) include methanol or other organic solvents? If not, the analysis should be carried in aqueous extract, not in methanol extract. There is a lack of information about how the bark paste is made of. 3.         What is the cell viability assay shown in Figure 7?? The pictures in Figure 7 are low quality and a quantitative analysis should be carried out and stats should be shown. It should be carried out in order to evaluate the cytotoxic nature of the B. alnoides bark extract compared t other species (see refs. 76 and 77). The authors indicated that MTT assay was performed but the data and analysis regarding the In vitro cytotoxicity assay were not shown. 4.         As indicated in Results, “Information on traditional phytotherapy practices were gathered from remote areas of Darjeeling district by interviewing the selected local informants” and “address and location of the residence of each participant was recorded using global positioning system (GPS)”. Geographical information should be provided for the different practices. Anyassociation between recorded medicinal plants and geographical spots in the district? 5. Lanes 370-374, 387-406 should be part of the Discussion, not part of the Results section.   Minor.   1.         "knowledgeable persons" Should be defined. (line 12) 2.         Missing space” ‘between two generationsis” (lanes 60, 200) 3.         Extra spaces (lines 112, 113, 196, 309, 346) 4.         “ nti-inflammatory” should be replaced to “ anti-inflammatory in Table 8 5.         Funding and Data Availability Statement sections are missing. 6.         References should be appropriated formatted.

Author Response

From,                                                                                                                          

Prof. Chowdhury Habibur Rahaman

Department of Botany

Visva-Bharati University

Santiniketan,West Bengal, India

e-mail: habibur_cr@visva-bharati.ac.in

To,

The Reviewer,

Plants, MDPI

Dear sir/madam,

I would like to thank you for this opportunity to resubmit a revised copy of our manuscript entitled “Quantitative ethnobotany of medicinal plants from Darjeeling District of West Bengal, India along with phytochemistry and toxicity study of BetulaalnoidesBuch.-Ham. ex D.Don bark” (Manuscript ID- 3142538). We, all the authors, would like to take this opportunity to express our gratitude for the positive feedback and insightful comments, which have greatly helped us to improve the quality of our manuscript.

In accordance to the comments, all changes in the text were marked up using the “Track Changes” function in the revised manuscript. Enclosed please find our revised manuscript and list of changes made point-by-point.

Hope our revised manuscript will satisfy you in all respect.

Regards,

Yours sincerely,

Prof. Chowdhury Habibur Rahaman

Reviewer 2

Major

  1.  Table 8 is not fully readeable.

Revised manuscript (RM) Table 8 (Page 63): Thank you for the suggestion. In the revised manuscript, the Table 8 has been corrected.

  1. Does the bark paste of this plant (topically applied for treating snakebite, inflammation, cut and wounds) include methanol or other organic solvents? If not, the analysis should be carried in aqueous extract, not in methanol extract. There is a lack of information about how the bark paste is made of?

- Thank you for your valuable comment. You are correct in pointing out that the preparation of bark paste involves water, whereas in our phytochemical analysis, we employed 100% methanol as solvent. We chose 100% methanol as the solvent due to its demonstrated efficiency in extracting greater amount of phenolic compounds than the water solvent. Several previous studies evidenced this fact of higher amount of phenolic extraction by the methanol. For instance, studies on the effect of extraction solvents on the total phenolic content (TPC) in the aerial parts of root vegetables showed that 100% methanol yielded significantly higher TPC. Onion, white radish, red radish, beetroot, and carrot demonstrated TPC values of 16.90, 29.59, 37.09, 31.73, and 66.33 mg GAE/g, respectively, when extracted with methanol, while aqueous extracts had much lower TPC values of 11.01, 12.31, 18.99, 20.78, and 9.59 mg GAE/g, respectively (https://doi.org/10.3390/agriculture12111820). In another study, 100% methanolic extract demonstrated significantly higher total phenolic content (TPC) from the Moringa oleifera leaf compared to aqueous extracts. The methanolic extract yielded 186.7 mg GAE/g, while the aqueous extract yielded only 78.2 mg GAE/g. ((https://doi.org/10.3390/molecules14062167).

Yes, there is a lack of information regarding preparation of bark paste of Betula alnoides. This information has been incorporated in the Table 6 against Betula alnoides.  The collected fresh barks of B. alnoides were washed and cut into small pieces. The small pieces of bark were then ground and made into paste with water using a mortar and pestle. The paste was applied topically to treat snakebite, inflammation and cuts.

  1. What is the cell viability assay shown in Figure 7?? The pictures in Figure 7 are low quality and a quantitative analysis should be carried out and stats should be shown. It should be carried out in order to evaluate the cytotoxic nature of the B. alnoidesbark extract compared t other species (see refs. 76 and 77).
  • What is the cell viability assay shown in Figure 7: Thank you for allowing us to provide this explanation. The cell viability was assessed using the MTT assay. Changes in cell morphology of L929 mouse fibroblast cells was observed before and after 24-hour exposure to different concentrations (50–300 mg/L) of the bark extract to evaluate the cytotoxic effects of the extract. The images illustrate the morphological differences between the treated and untreated cells. The caption has been changed as: Phase Contrast micrographs showing the morphological changes in the L929 mouse fibroblast cells before and after 24 h exposure to different concentrations (50–300 mg/L) of B. alnoides bark extract, along with negative control.
  • The pictures in Figure 7 are low quality: We acknowledge the concern regarding the picture quality in Figure 7. In response, we have provided a high resolution image of Figure 7 in a separate ZIP file.  
  • A quantitative analysis should be carried out and stats should be shown: In accordance to your suggestion the result section of the toxicity study has been modified (RM line no. 372-381).
  • The authors indicated that MTT assay was performed but the data and analysis regarding the In vitro cytotoxicity assay were not shown: According to your suggestion, all the data of the in vitro cytotoxicity assay has been provided in Supplementary Table 2 for further reference.
  1. As indicated in Results, “Information on traditional phytotherapy practices were gathered from remote areas of Darjeeling district by interviewing the selected local informants” and “address and location of the residence of each participant was recorded using global positioning system (GPS)”. Geographical information should be provided for the different practices. Any association between recorded medicinal plants and geographical spots in the district?

-Thank you sir/madam, for allowing us to explain your query.Geographical information of each informant’s residence and plant collection area have provided in the Supplementary Table 1. Collection site for each of the medicinal plants was not recorded, but the locations of the collection areas from where medicinal plants were collected have been noted using the GPS. Some medicinal plants were collected in the home gardens which was near the residential area and most of the plants were collected from nearby forest.

5.Lanes 370-374, 387-406 should be part of the Discussion, not part of the Results section. 

[Revised manuscript (RM) Lines 499-504, 526-536]: The lines are already in the discussion part.

Minor.  

1."knowledgeable persons" Should be defined. (line 12)

Thank you for allowing us to explain about “knowledgeable person”. In this article, the ethnobotanical data collection was carried out using the purposive sampling method to identify 47 specialist informants (key informant and knowledgeable persons). Key informants are mainly traditional herbalists who practise ethnomedicine for treating different kinds of diseases. Knowledgeable persons are people who have much more ethnomedicinal knowledge than other people of the community.

  1. Missing space” ‘between two generationsis” (lanes 60, 200).

Revised manuscript (RM) Lines 61, 240: Thank you sir/ma’am for pointing out this. A gap has been placed in between “generation” and “is”.

  1. Extra spaces (lines 112, 113, 196, 309, 346)

Revised manuscript (RM)Lines 114, 115, 233, 437, 475: The extra spaces have deleted.

  1. “nti-inflammatory” should be replaced to “anti-inflammatory in Table 8.

Revised manuscript (RM) Table 8: Thank you sir/ma’am for pointing this out. The word “nti-inflammatory” is replaced to “anti-inflammatory” in the Table 8.

  1. Funding and Data Availability Statement sections are missing.

Revised manuscript (RM) Line no. 982-983: The funding and data availability statement sections have been corrected.

  1. References should be appropriated formatted.

Revised manuscript (RM):Thank you sir/ma’am for the suggestion. References have been formatted strictly following to the guidelines provided in the journal.

Reviewer 3 Report

Comments and Suggestions for Authors

I have 4 comments for the Authors:

1. How did you obtain ethnomedicinal information, by semi-structured interviewing selected herbalists and 36 knowledgeable persons? It was found that the Fic value of medicinal plants was mostly used for the category of digestive diseases, as an antioxidant and anti-inflammatory agent, the phenolics affect other biological systems and diseases as well. This insufficiency of information should be considered.

2. What is the wide array of phototherapeutic 98 traditionally used by the folk communities residing in rural pockets of the district Darjeeling to elucidate the phytochemical and toxicity profiles of methanolic bark extract of 100 B.  Alnoides. This should take an amount of time and teamwork.  What kind of standard technique was used?

3. Similar articles relating to the study of medicinal plants from Darjeeling District of West Bengal, phytochemistry, and toxicity, what are your differences in your article from other similar papers? 

4 The bark extract contains significant amounts of phenolics and tannin compounds
have been reviewed and published.
The leaves and fruits should be better used as the source of phenolics. It will be easier for collection and extraction and the tree conservation.
This needs to be mentioned and supported.

Comments on the Quality of English Language

English is understandable except for some terms.

Author Response

From,                                                                                                                          

Prof. Chowdhury Habibur Rahaman

Department of Botany

Visva-Bharati University

Santiniketan,West Bengal, India

e-mail: habibur_cr@visva-bharati.ac.in

To,

The Reviewer,

Plants, MDPI.

Dear sir/madam,

I would like to thank you for this opportunity to resubmit a revised copy of our manuscriptentitled “Quantitative ethnobotany of medicinal plants from Darjeeling District of West Bengal, India along with phytochemistry and toxicity study of Betula alnoidesBuch.-Ham. exD.Don bark” (ManuscriptID- Plants-3142538). We, all the authors, would like to take this opportunity to express our gratitude for the positive feedback and insightful comments, which have greatly helped us to improve the quality of our manuscript.

In accordance to the comments, all the changes in the text were marked up using the “Track Changes” function in the revised manuscript. Enclosed please find our revised manuscript and a word file containing our point-by-point responses.

Hope the present form of the revised manuscript will satisfy you in all respect.

Regards,

Yours sincerely,

Prof. Chowdhury Habibur Rahaman

  1. How did you obtain ethnomedicinal information, by semi-structured interviewing selected herbalists and 36 knowledgeable persons?

- Thank you sir/madam, for allowing us to explain our process of data collection. In this article, the ethnobotanical data collection was carried out using the purposive sampling method to identify 47specialist informants (key informant/knowledgeable persons) who have much more ethnomedicinal knowledge than other people of the community. We then conducted semi-structured interviews using open-ended questions to gather detailed information, followed by a free listing after clearly presenting the purpose of the study and its outcome before the participants.

  1. It was found that the Fic value of medicinal plants was mostly used for the category of digestive diseases, as an antioxidant and anti-inflammatory agent, the phenolics affect other biological systems and diseases as well. This insufficiency of information should be considered.

- Thank you for highlighting this. According to your suggestion we have revised the discussion to explain the significance of phytochemicals in relation to the traditional uses of 23 medicinal plants enlisted under the digestive system disorders category. Through a critical literature review we attempted to correlate the traditional uses of the listed plants with their pharmacological potentials. The discussion made on the aforementioned area has incorporated in our Revised Manuscript (Line No. 497-534). 

  1. What is the wide array of phototherapeutic 98 traditionally used by the folk communities residing in rural pockets of the district Darjeeling to elucidate the phytochemical and toxicity profiles of methanolic bark extract of 100 B.  alnoides. This should take an amount of time and teamwork.  What kind of standard technique was used?

- Yes sir/madam, this comprehensive research required time, effort, and teamwork.

The plant Betula alnoides has been selected for its phytochemical and toxicity studies using a quantitative tool namely the Fidelity Level (%) index. The plant scored the highest percentage (97%) in Fidelity Level among the recorded species. The fidelity level (%) is an index commonly used in ethnobotanical data analysis to identify the plants with therapeutic potential, making them promising candidates for further scientific exploitation of their phytochemical as well as pharmacological properties. Our investigation explored the reliance of local informants upon the B. alnoides used frequently for treatments of snake bites, wound healing, and inflammation. Our approach has been holistic, beginning with ethnomedicinal knowledge, we advanced to phytochemical and toxicity assessments, aiming to bridge traditional wisdom with scientific validation.

  1. Similar articles relating to the study of medicinal plants from Darjeeling District of West Bengal, phytochemistry, and toxicity, what are your differences in your article from other similar papers?

-Sir/Madam, you have correctly pointed out that the Eastern Himalayan region, including the Darjeeling district, has attracted significant attention from researchers who have documented the rich herbal knowledge sustained by its diverse ethnic communities. As per your suggestion, the novelty of our research work has been illustrated in the revised manuscript comparing earlier studies on ethnobotany (Line No. 547-557), phytochemistry (Line No. 559-566) and toxicity (Line No. 633-635) of medicinal plants from Darjeeling district.

  1. The bark extract contains significant amounts of phenolics and tannin compounds have been reviewed and published. The leaves and fruits should be better used as the source of phenolics. It will be easier for collection and extraction and the tree conservation. This needs to be mentioned and supported.

Thank you for the opportunity to explain these aspects. Please refer to lines 612-628 and lines 845-860 in the revised manuscript for explanations made following your comments.

Round 2

Reviewer 1 Report

Comments and Suggestions for Authors

Dear sir,

the article 'Quantitative ethnobotany of medicinal plants from Darjeeling 2 District of West Bengal, India along with phytochemistry and 3 toxicity study of Betula alnoidesBuch.-Ham. ex D.Don bark' has improved its quality after revision. However, many problems with spacing show up throughout the manuscript. It should be corrected, despite many could appear after conversion to .pdf, but not all. I have downloaded an annotated file with corrections.

It is a long and interesting article describing medicinal applications of plants in north east India. The research is more horizontal and descriptive, than vertical and deepening. The Betula study is also interesting.

Best regards

Comments on the Quality of English Language

Dear sir,

the article 'Quantitative ethnobotany of medicinal plants from Darjeeling 2 District of West Bengal, India along with phytochemistry and 3 toxicity study of Betula alnoidesBuch.-Ham. ex D.Don bark' has improved its quality after revision. However, many problems with spacing show up throughout the manuscript. It should be corrected, despite many could appear after conversion to .pdf, but not all. I have downloaded an annotated file with corrections.

It is a long and interesting article describing medicinal applications of plants in north east India. The research is more horizontal and descriptive, than vertical and deepening. The Betula study is also interesting.

Best regards

Author Response

From,                                                                                                                          

Prof. Chowdhury HabiburRahaman

Department of Botany

Visva-Bharati University

Santiniketan,West Bengal, India

e-mail: habibur_cr@visva-bharati.ac.in

To,

The Reviewer,

Plants, MDPI

Dear sir/madam,

I would like to thank you for this opportunity to resubmit a revised copy of our manuscript entitled “Quantitative ethnobotany of medicinal plants from Darjeeling District of West Bengal, India along with phytochemistry and toxicity study of Betula alnoidesBuch.-Ham. ex D.Don bark” (Manuscript ID- 3142538). We, all the authors, would like to take this opportunity to thank you for your kind remarks and encouraging feedback. Your positive response has motivated us to work harder for future.According to your suggestion, we have carefully reviewed and corrected the manuscript.

Regards,

Yours sincerely,

Prof. Chowdhury Habibur Rahaman

Reviewer 3 Report

Comments and Suggestions for Authors

This article is quite a large interview surveying  115 species with over 65 families and 104 genera., a total of 101 monoherbal- and 21 polyherbal formulations that  65 families of medicinal and indigenous plants from Darjeeling District, India. Congratulations for a nice work.

My comments:

1. The data collection and analysis can be used as an upgraded database of Darjeeling's medicinal plants. In  future, these valuable medicinal herbs may be further investigated and applied as potential sources for modern drugs, health-promoting products, and cosmetic discovery. Should the authors not mention, criticize, and discuss these in the paper as well?

2  Update the article: There are several interesting studies about B. alnoides to be added to the Discussion and References, such as :

"Evaluation of Anti-HIV-1 Integrase and Anti-Inflammatory Activities of Compounds from Betula alnoides Buch-Ham

 https://doi.org/10.1155/2019/25739

"Antioxidant, antimicrobial activity and inhibition of α-glucosidase activity by Betula alnoides Buch. bark extract and their relationship with polyphenolic compounds concentration"

doi: 10.3109/08923973.2012.661739.

And more papers concerned with B. alnoides should be added..

  1.    
  2.  

Comments on the Quality of English Language

Some few words to be approved are already  shown in the full manuscript

Author Response

From,                                                                                                                          

Prof. Chowdhury Habibur Rahaman

Department of Botany

Visva-Bharati University

Santiniketan,West Bengal, India

e-mail: habibur_cr@visva-bharati.ac.in

To,

The Reviewer,

Plants, MDPI

Dear sir/madam,

I would like to thank you for this opportunity to resubmit a revised copy of our manuscriptentitled “Quantitative ethnobotany of medicinal plants from Darjeeling District of West Bengal, India along with phytochemistry and toxicity study of Betula alnoides Buch.-Ham. ex D.Don bark” (Manuscript ID- 3142538). We, all the authors, would like to take this opportunity to express our gratitude for the positive feedback and insightful comments, which have greatly helped us to improve the quality of our manuscript.

In accordance to the comments, all changes in the text were marked up using the “Track Changes” function in the revised manuscript. Enclosed please find our revised manuscript and list of changes made point-by-point.

Hope our revised manuscript will satisfy you in all respect.

Regards,

Yours sincerely,

Prof. Chowdhury Habibur Rahaman

  1. The data collection and analysis can be used as an upgraded database of Darjeeling's medicinal plants. In future, these valuable medicinal herbs may be further investigated and applied as potential sources for modern drugs, health-promoting products, and cosmetic discovery. Should the authors not mention, criticize, and discuss these in the paper as well?

- As per your suggestion, we have added a discussion highlighting the significance of ethnomedicinal plant wealth of the Darjeeling district and they can be further exploited for drug development and natural product innovation. Please refer to lines 623-658 of the revised manuscript for the detailed discussion.

  1. Update the article: There are several interesting studies about B. alnoides to be added to the Discussion and References, such as :

"Evaluation of Anti-HIV-1 Integrase and Anti-Inflammatory Activities of Compounds from BetulaalnoidesBuch-Ham

 https://doi.org/10.1155/2019/25739

"Antioxidant, antimicrobial activity and inhibition of α-glucosidase activity by BetulaalnoidesBuch. bark extract and their relationship with polyphenolic compounds concentration"

doi: 10.3109/08923973.2012.661739.

-In response to your feedback, we have incorporated a brief discussion in the revised manuscript on the biological activity studies of Betula alnoides highlighting its future prospect of varied range of bioactivities. You can find this discussion in lines 714-723 of the revised manuscript.

Round 3

Reviewer 1 Report

Comments and Suggestions for Authors

Dear sir, the article is now ready for publication.

Best

Comments on the Quality of English Language

Dear sir, the article is now ready for publication.

Best

Author Response

From,                                                                                                                          

Prof. Chowdhury Habibur Rahaman

Department of Botany

Visva-Bharati University

Santiniketan,West Bengal, India.

e-mail: habibur_cr@visva-bharati.ac.in

To

The Reviewer,

Plants, MDPI.

Dear sir/madam,

I want to express my heartfelt gratitude for your thoughtful review of our manuscript entitled “Quantitative ethnobotany of medicinal plants from Darjeeling District of West Bengal, India, along with phytochemistry and toxicity study of Betula alnoidesBuch.-Ham. ex D.Don bark” (Manuscript ID- 3142538). Your positive feedback and insights greatly enhanced the quality of our work, and I appreciate the time you invested in reviewing our manuscript.

Thank you once again for your support.

Regards,

Yours sincerely,

 Prof. Chowdhury Habibur Rahaman

Reviewer 3 Report

Comments and Suggestions for Authors

I agree with other 2 reviewers.

Author Response

From,                                                                                                                          

Prof. Chowdhury Habibur Rahaman

Department of Botany

Visva-Bharati University

Santiniketan,West Bengal, India.

e-mail: habibur_cr@visva-bharati.ac.in

To

The Reviewer,

Plants, MDPI.

Dear sir/madam,

I am writing to express my sincere appreciation for your thoughtful review of our manuscript, "Quantitative Ethnobotany of Medicinal Plants from Darjeeling District of West Bengal, India, along with Phytochemistry and Toxicity Study of Betula alnoidesBuch.-Ham. ex D.Don Bark.". Your expert feedback and suggestions significantly improved our work, and I am grateful for the time and effort you dedicated to reviewing our research.

Thank you again for your valuable contribution.

Regards,

Yours sincerely,

 Prof. Chowdhury Habibur Rahaman
